# THE CHOSEN FEW: SPARSE ADAPTATION FOR LARGE MODELS

## ABSTRACT

Parameter-Efficient Fine-Tuning (PEFT) methods have become essential for adapting large pretrained models to downstream tasks, with Low-Rank Adaptation (LoRA) emerging as one of the most widely adopted solutions. However, there remain several key limitations in current LoRA-based PEFT methods: (1) the low-rank feature space in LoRA is rigid, reducing its capacity for dynamic adaptation; (2) the restricted dimensionality, coupled with dense and entangled representations, constrains the model's capacity to generalize across multiple domains; and (3) the compression process limits the extent to which model behavior can be understood from the learned representations, making it difficult to interpret the functional role of task-relevant features. In this paper, we argue that *sparse adaptation* offers a principled and more flexible alternative to low-rank adaptation, with the added benefit of enhancing interpretability. Instead of compressing information into a low-rank subspace, sparse adaptation focuses on identifying and selectively activating a small subset of high-dimensional latent features, enabling a more decomposed and dynamic fine-tuning process. Building on this paradigm, we propose STAN (**S**parse adap**TA**tio**N**), a novel method that actualizes sparse adaptation by integrating dedicated Sparse Autoencoder (SAE) modules into frozen pretrained models. STAN learns to encode task-specific adaptations through sparse activations within the SAEs, thereby using sparse features as the mechanism for dynamic and robust adaptation. Beyond the flexibility offered by input-dependent sparse combinations, the large latent space of the SAEs provides scalable capacity for cross-domain adaptation, while their inherent semantic decomposition structure supports more interpretable representations. Through extensive experiments, we demonstrate that STAN outperforms state-of-the-art PEFT baselines across a range of benchmarks, while uniquely enabling inspection and analysis of the learned sparse activations. Our findings position sparse adaptation as a promising new direction in PEFT, advancing both the expressivity and interpretability of model adaptation.

## 1 INTRODUCTION

Large-scale pretrained models have demonstrated strong generalization across a wide range of modalities, including language, vision, and multimodal tasks. Representative examples include large language models (LLMs) (Bai et al., 2023; Brown et al., 2020; Guo et al., 2025), vision-language models (VLMs) (Liu et al., 2023b), diffusion-based image generators (Ho et al., 2020; Rombach et al., 2022), and large vision transformers (Dosovitskiy et al., 2020), which have achieved state-of-the-art zero-shot and few-shot performance in a wide range of downstream tasks (Kojima et al., 2022). These capabilities are largely attributed to pretraining on massive and heterogeneous datasets, enabling models to internalize broad statistical regularities and encode diverse semantic knowledge. Despite their versatility, adapting such large foundation models, often containing billions of parameters, to new tasks or domains requires substantial computational resources and specialized expertise. Full model fine-tuning typically requires extensive task-specific supervision, incurs significant computational cost, and can lead to undesirable side effects such as catastrophic forgetting (Luo et al., 2023) and memorization of sensitive data (Carlini et al., 2019). These challenges have motivated the development of parameter-efficient fine-tuning (PEFT) methods (Houlsby et al., 2019; Hu et al., 2022; Li & Liang, 2021; Xu et al., 2023), which aim to adapt models by modifying only a small subset of parameters while maintaining performance.

Among PEFT methods (Houlsby et al., 2019; Hu et al., 2022; Li & Liang, 2021; Xu et al., 2023), Low-Rank Adaptation (LoRA) (Hu et al., 2022) has gained widespread adoption due to its simplicity and strong empirical performance. By inserting trainable low-rank matrices into frozen pretrained layers, LoRA enables efficient adaptation with minimal overhead, often achieving results comparable to full model fine-tuning across a range of downstream tasks. However, LoRA's adaptation mechanism introduces several key limitations: (1) The low-rank feature space inherent in LoRA (Hu et al., 2022) imposes structural rigidity, significantly limiting the model's capacity for dynamic adaptation to diverse data distributions (Wang & Zhao, 2025). This inflexibility hinders the alignment of adapted features with the specific and evolving demands of heterogeneous tasks and data characteristics. (2) The restricted dimensionality, often coupled with the emergence of dense and entangled representations, substantially constrains the model's ability to generalize across distinct domains (Zhang et al., 2025). The lack of clear separability in these compressed features diminishes the model's capacity to learn domain-specific nuances and adapt meaningfully in multi-domain or cross-distribution contexts. (3) The inherent compression in LoRA's mechanism reduces the interpretability of the learned representations (Nijasure et al., 2025). This opacity makes it difficult to interpret the functional roles of task-relevant features, limiting our ability to identify which components drive adaptation, and thus posing a challenge to understanding model behaviors. These limitations collectively lead to reduced adaptability, insufficient domain generalization, and limited capacity to revise, debug, or exert fine-grained control over the adaptation process. In addition, the limited understanding of model behavior (Chen et al., 2025) falls short of addressing the growing demand for *interpretable adaptation* in large models – a need that has been increasingly emphasized in recent work (Mumuni & Mumuni, 2025; Wang et al., 2025).

To address the gap above, we propose a shift from *low-rank compression* to *sparse adaptation* – a framework grounded in sparse feature learning (Cunningham et al., 2023; Olshausen & Field, 1997). Rather than encoding adaptation into rigid, dense, and polysemantic subspaces, STAN seeks to *dynamically* activate a sparse subset of high-dimensional, disentangled features that more directly reflect task-specific changes in model behavior. Sparse Autoencoders (SAEs) (Cunningham et al., 2023; Makhzani & Frey, 2013) provide a natural mechanism for this goal, enabling models to learn robust representations with sparsity constraints that promote structure separability and semantic decoupling. Furthermore, the broad representation capacity and the sparsity rendered by SAEs can provide the adaptation with more selection combinations, which not only enhances the *dynamics* of fine-tuning but also strengthens the model's ability to generalize across diverse domains.

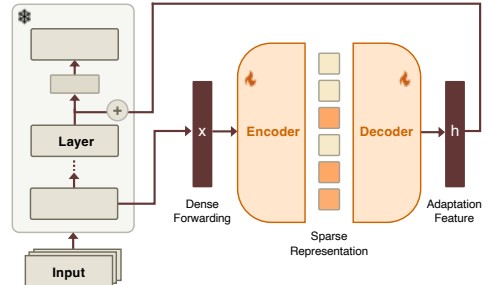

Figure 1: **Overview of STAN pipeline**. $x$ is the dense forwarding and $h$ denotes the adaptation feature to the next layer. The layer block is a frozen pretrained model layer. The neuron between the Encoder and Decoder is latent feature, and only the top $K$ activations (orange) are passed to the decoder, while the rest are masked out.

We instantiate this framework with STAN (**S**parse adap**TA**tio**N**), a new PEFT method designed to enhance the efficiency and dynamics of adaptation, expand the overall representational capacity, and enable flexible, scalable adaptation with improved interpretability. Rather than projecting into low-rank dense subspaces, STAN integrates lightweight SAE modules into selected layers of a frozen pretrained model. These modules encode adaptation signals using sparsely activated high-dimensional features, offering a richer representational space with only modest overhead addition. STAN achieves adaptation efficiency comparable to LoRA while enabling more scale feasible feature space selection and more broad representation capacity over the adaptation process, facilitating the adaptation process to capture the diverse decomposed information in multi-space combination. We validate STAN across a diverse range of tasks and architectures, demonstrating that it achieves competitive performance with strong PEFT baselines. Moreover, besides the reasoning tasks, we extend STAN to diffusion-based generative models (Ho et al., 2020; Rombach et al., 2022), where we show that the learned sparse features support qualitative and quantitative analysis, enabling structured interventions such as multi-style alignment and interpretability with disentangled features. An overview of STAN's architecture is provided in Figure 1. We summarize our main contributions as follows:

- We introduce STAN, a novel PEFT method that instantiates the Sparse Adaptation paradigm using Sparse Autoencoders, offering a conceptually distinct alternative to low-rank adaptation methods such as LoRA, and enabling a more dynamic and flexible fine-tuning process.

- We demonstrate the effectiveness of STAN across a range of tasks and model architectures spanning multiple modalities, showing that it achieves performance superior to state-of-the-art baselines. Empirical analysis further reveals that STAN retains adaptation stability while improving the dynamics of the fine-tuning process through the aid of sparsity.

- We extend STAN to a state-of-the-art diffusion-based generative model (Rombach et al., 2022) in the context of a style alignment task, showcasing its versatility and further highlighting its large representational capacity to capture and isolate diverse styles. We aim to enhance interpretability by dynamically decomposing dense features into sparse, disentangled components. By understanding the semantics of each component and their interactions, we hope to be able to reason about the behavior of the entire adaptation mechanism.

## 2 RELATED WORK

**Parameter-Efficient Fine-Tuning Methods.**    Parameter-Efficient Fine-Tuning has emerged as an effective strategy for adapting large pretrained models to downstream tasks through dedicated modifications (Houlsby et al., 2019). Rather than updating all model parameters, PEFT focuses on adjusting a small subset, significantly reducing computational and data requirements. PEFT methods can be broadly categorized into three classes (Prottasha et al., 2025): prompt tuning, adapter modules, and low-rank decomposition adaptation methods: (i) prompt tuning (Lester et al., 2021; Liu et al., 2023b; 2024a; Shen et al., 2024a) steers the model outputs by optimizing either continuous or discrete prompts while keeping the backbone frozen. These prompts are typically injected into the input or intermediate layers and act as virtual tokens that guide the model's predictions. A prominent example is Prefix-Tuning (Li & Liang, 2021), which prepends trainable vectors into the attention mechanism, enabling effective task adaptation with minimal parameter updates; (ii) Adapter modules (Houlsby et al., 2019; He et al., 2021; Hu et al., 2023) introduce additional lightweight layers within pretrained architectures. These modules are often structured as bottlenecks, comprising a down-projection, non-linear activation, and an up-projection, allowing the core model to remain untouched. AdapterHub (Pfeiffer et al., 2020) exemplifies this modularity, supporting plug-and-play adapters across tasks; (iii) *Low-rank decomposition adaptation methods* (Hu et al., 2022; Liu et al., 2023c; Tian et al., 2024; Xia et al., 2024) take a different approach by approximating weight updates using low-rank matrices. LoRA (Hu et al., 2022) is the canonical method, freezes the base model and injects trainable low-rank matrices into specific layers. Several extensions have since been proposed: AdaLoRA (Zhang et al., 2023) dynamically allocates parameter budgets based on learned importance scores; SoRA (Ding et al., 2023) modulates the intrinsic rank during fine-tuning through proximal gradient-based gating; and HydraLoRA (Tian et al., 2024) introduces an asymmetric architecture that removes the need for task-specific tuning expertise in complex scenarios. Other works like (He et al., 2023a; Zhang et al., 2024b; Fu et al., 2022; Shen et al., 2024b), they focus on selecting or sparsifying subsets of existing parameters rather than learning sparse latent features.

Despite their efficiency, current PEFT approaches, whether using auxiliary modules as in prompt tuning and adapters, or low-rank approximations as in LoRA, share a common drawback: they encode adaptations in dense and entangled representations. This results in insufficient dynamics during the fine-tuning process, constrained representational capacity for multi-domain adaptation, and limited interpretability. As a result, there is growing interest in developing advanced PEFT methods that retain strong adaptation performance while introducing *structured sparsity*, with the goal of enhancing *fine-tuning dynamics*, *representational flexibility*, and *interpretability*.

**Sparse Autoencoder.**    Sparse autoencoders (SAEs) (Ng et al., 2011; Makhzani & Frey, 2013) provide a scalable, unsupervised framework for learning compact and disentangled representations by imposing sparsity constraints on hidden activations. This methodology has recently gained substantial attention for its ability to enhance the interpretability of foundation models, including large language models (LLMs) (Cunningham et al., 2023; Templeton et al., 2024), vision-language models (VLMs) (Zhang et al., 2024a), and CLIP-style architectures (Lim et al., 2024). A core idea of SAEs lies in learning to reconstruct inputs using a sparse set of features in a higher-dimensional space, effectively disentangling superposed features (Bricken et al., 2023; Shi et al., 2025).

Specifically, given a data input $x \in \mathbb{R}^n$, encoder $E \in \mathbb{R}^{l \times n}$, and decoder $D \in \mathbb{R}^{n \times l}$, the autoencoding process of a $\mathrm{topk}$ SAE (Gao et al., 2024) can be formulated as:

$$z_k := \mathrm{topk}(Ex), \quad \hat{x} := D z_k,$$

where $\mathrm{topk}(\cdot)$ selects the top $k$ ($k \ll l$) largest elements of the input and zeroes out the remaining values. This enforced sparsity constraint facilitates the emergence of semantically meaningful representations $z_k$, effectively disentangling the complex, superposed features within large models.

Alongside interpretability, the potential of SAEs in other areas has also been explored. In efficiency optimization, SAEs demonstrate remarkable versatility by addressing both retrieval and computational constraints through sparse coding. Recent works (Kim et al., 2024; Wen et al., 2025) exemplify this capability: the former optimizes the tradeoff between retrieval accuracy and efficiency through sparse contrastive learning and sparse matrix factorization, while the latter compresses LLM key-value caches using universal dictionaries. These approaches transform dense, high-dimensional representations into sparse, efficient formats that preserve essential information while largely reducing computational overheads.

In the context of safety alignment and controllable generation in diffusion models (Cywiński & Deja, 2025; Kim & Ghadiyaram, 2025; Shi et al., 2025; Surkov et al., 2024; Tian et al., 2025), SAEs are implemented as zero-shot concept detectors, enabling precise identification of both desired and undesired features during the generative process. By isolating specific semantic features in the model's latent space, SAEs facilitate targeted interventions while maintaining the overall quality of generation. Although SAEs have been adapted to a wide range of tasks and domains, their benefits have been largely overlooked in the context of PEFT. This presents a significant research gap: integrating the representational capabilities of SAEs with PEFT techniques can not only expand the adaptation space, but also enable more flexible feature selection and improve interpretability. In addition, the sparse structure of SAEs introduces greater dynamism into the adaptation process, mitigating the rigidity typically associated with the low-rank spaces.

## 3 METHODOLOGY

**Background.** Parameter-efficient fine-tuning has become a key paradigm for adapting large-scale pretrained foundation models to downstream tasks without incurring the full computational cost of end-to-end fine-tuning (Xu et al., 2023). Among various PEFT methods, LoRA (Hu et al., 2022) has emerged as one of the most widely used due to its simplicity and empirical effectiveness. Instead of modifying the original weights directly, LoRA introduces a low-rank trainable update that captures task-specific information.

Specifically, given a layer with a pretrained weight matrix $W_0 \in \mathbb{R}^{m \times n}$, LoRA models the task-specific adaptation as a low-rank update to the weights:

$$\Delta W = \frac{\alpha}{r} BA, \tag{1}$$

where $A \in \mathbb{R}^{r \times n}$ and $B \in \mathbb{R}^{m \times r}$ are low-rank matrices with $r \ll \min(m, n)$, and $\alpha$ is a scaling factor. The forward pass through the adapted layer, receiving input $x \in \mathbb{R}^n$ and producing output $h \in \mathbb{R}^m$, is then modified as follows:

$$h = W_0 x + \Delta W x. \tag{2}$$

Here, $W_0$ remains frozen, and only the components representing $\Delta W$ are updated during training. The core idea of LoRA lies in constraining the update matrix $\Delta W$ to possess a low intrinsic rank, denoted by $r$. While this low-rank adaptation strategy is highly effective in terms of performance and efficiency, compression of the task adaptation into dense, low-dimensional subspace defined by matrices $A$ and $B$ inherently leads to the entanglement of potentially distinct underlying concepts, posing significant challenges for interpretability and mechanistic understanding of learned adaptation.

However, while effective, LoRA has inherent limitations rooted in its architectural constraints. The low-rank structure of the matrices $A$ and $B$ severely restricts the representation capacity of the model (Wang & Zhao, 2025; Zhang et al., 2025), and the compression nature of LoRA reduces its ability to capture *dynamic* patterns during parameter updates. This lack of adaptability may lead to information obfuscation and hinders the model's capacity to generalize across heterogeneous or

multidisciplinary domains. Furthermore, prior studies (Ding et al., 2023; He et al., 2025; Liu et al., 2023c; Mao et al., 2025; Zhang et al., 2023) have noted that compressing adaptations into a rigid low-rank subspace can result in a representational bottleneck, limiting the expressivity required for complex task-specific transformations. In addition, the dense structure of the low-rank projections often entangles distinct adaptation features, making it difficult to assign functional meaning to individual components within the update matrix $\Delta W$. These limitations underscore the necessity for a new PEFT paradigm that enables more adaptive and flexible representations while enabling dynamic, structured, and interpretable adaptation.

### 3.1 SPARSE ADAPTATION

To this end, we propose STAN, a high-dimensional, dynamic sparse adaptation framework, inspired by Sparse Autoencoders (Cunningham et al., 2023). It is designed to provide broader representational capacity and dynamic feature selection throughout the adaptation process. By introducing sparsity, STAN enhances the dynamics of the adaptation process, equipping the model with the capability to generalize across multiple domains and offering improved potential for interpretability.

Similar to LoRA, STAN modifies the forward pass by adding a learned adaptation $\Delta W x$ to the frozen pretrained output $W_0 x$, as in Eq. 2, but changes how this adaptation is computed. Specifically, STAN introduces an encoder matrix $E \in \mathbb{R}^{l \times n}$ and a decoder matrix $D \in \mathbb{R}^{m \times l}$, where $l$ defines the dimension of a high-dimensional latent space. The input $x$ is first projected into this space via $E$, and sparsity is enforced using a $\mathrm{topk}$ operator that selects only the top $k$ activations, zeroing out the rest. The sparse code is then decoded by $D$ to produce the adaptation signal:

$$\Delta W x = \frac{1}{k} D \cdot \mathrm{topk}(Ex). \tag{3}$$

The complete forward pass becomes:

$$h = W_0 x + \frac{1}{k} D \cdot \mathrm{topk}(Ex), \tag{4}$$

where $k \ll l$ is a sparsity-controlling hyperparameter. Only $E$ and $D$ are updated during training, while $W_0$ remains frozen.

### 3.2 STAN AS A NON-LINEAR COMBINATION OF SUBSPACES

A **key distinction** between STAN and LoRA lies in the fact that STAN's adaptation $\Delta W x$ is a nonlinear, input-dependent function. This nonlinearity stems from the $\mathrm{topk}$ operator, which *dynamically* selects a different subset of latent dimensions depending on $x$.

Let $\mathcal{I}(x) \subset 1, \ldots, l$ denote the indices of the top-$k$ activations for a given input $x$. We define an input-dependent selection matrix $S(x) \in 0, 1^{l \times l}$ as a diagonal matrix with $S(x)_{ii} = 1$ if $i \in \mathcal{I}(x)$ and 0 otherwise. This allows the sparsity operation to be expressed as $\mathrm{topk}(Ax) = S(x)Ax$. Substituting into Equation 3 yields:

$$\Delta W x = DS(x)Ex. \tag{5}$$

Equivalently, let $E_{\mathcal{I}(x)} \in \mathbb{R}^{k \times n}$ denote the submatrix of $E$ consisting of the rows indexed by $\mathcal{I}(x)$, and let $D_{\mathcal{I}(x)} \in \mathbb{R}^{m \times k}$ denote the submatrix of $D$ consisting of the corresponding columns. Then, for a given input $x$, the adaptation is equivalent to a projection onto a $k$-dimensional subspace defined by the selected latent features:

$$\Delta W x = D_{\mathcal{I}(x)} E_{\mathcal{I}(x)} x. \tag{6}$$

Since the index set $\mathcal{I}(x)$ varies with the input $x$, STAN effectively operates over a mixture of input-dependent $k$-dimensional subspaces, each spanned by a distinct subset of the $l$-dimensional latent feature space. The $\mathrm{topk}$ operator functions as a dynamic, non-linear router, selecting the most relevant features for each input. This mechanism enables STAN to model a richer class of adaptation functions than LoRA, which is confined to a single, fixed low-rank subspace. By *dynamically* composing basis vectors from a larger representational space, STAN offers greater expressivity while maintaining sparsity. Furthermore, in contrast to LoRA, where increasing the rank results in an exponential growth in trainable parameters, the large representational capacity offered by STAN allows for *flexible scaling* of feature selection without increasing the overall number of training parameters. For more discussion with related methods, see Appendix B.

In summary, the **core** of STAN is to leverage a non-linear, input-dependent sparse activation mechanism within a high-dimensional latent space to *dynamically* compose multiple adaptation subspaces, aiming for both representational flexibility and interpretability through sparsity. For more comprehensive discussion towards the interpretability and identifiability of STAN, please refer to Appendix A.

| Model | Method | Accuracy ↑ | | | | Matthew's Corr. ↑ |
|---|---|---|---|---|---|---|
| | | MNLI | SST-2 | QNLI | QQP | CoLA |
| RoBERTa-base | LoRA | 0.8514 | 0.9177 | 0.9177 | 0.8627 | 0.5981 |
| | AdaLoRA | 0.8429 | 0.9358 | 0.9225 | 0.8812 | 0.6132 |
| | SoRA | 0.7657 | 0.9220 | 0.8380 | 0.8420 | 0.5485 |
| | STAN (Ours) | **0.9303** | **0.9495** | **0.9408** | **0.9242** | **0.6191** |
| RoBERTa-large | LoRA | 0.8812 | 0.9553 | 0.9131 | 0.8842 | 0.6749 |
| | AdaLoRA | 0.8857 | 0.9472 | 0.9400 | 0.8883 | 0.6314 |
| | SoRA | 0.8769 | 0.9280 | 0.4860 | 0.8450 | 0.3470 |
| | STAN (Ours) | **0.8919** | **0.9610** | **0.9489** | **0.8957** | **0.7400** |
| DeBERTaV3-base | LoRA | 0.8857 | 0.9438 | 0.9371 | 0.9163 | 0.6729 |
| | AdaLoRA | 0.8637 | 0.9553 | 0.9440 | 0.8952 | 0.6864 |
| | SoRA | 0.8095 | 0.9564 | 0.9322 | 0.8540 | 0.6698 |
| | STAN (Ours) | **0.8974** | **0.9622** | **0.9477** | **0.9230** | **0.6904** |
| DeBERTaV3-large | LoRA | 0.8879 | 0.9599 | 0.9503 | 0.8923 | 0.7237 |
| | AdaLoRA | 0.9021 | 0.9587 | 0.9552 | 0.8899 | 0.7008 |
| | SoRA | 0.9056 | 0.9370 | 0.9440 | 0.8640 | 0.6829 |
| | STAN (Ours) | **0.9145** | **0.9622** | **0.9590** | **0.9058** | **0.7528** |

Table 1: Performance comparison on language understanding tasks across four large pretrained language models using five tasks from the GLUE benchmark. ↑ indicates that higher values are better. The best results are highlighted in **bold**.

| Method | QNLI | MNLI | SST-2 | QQP | MRPC | RTE | STSB |
|---|---|---|---|---|---|---|---|
| LoRA | 0.9371 | 0.8857 | 0.9438 | 0.9163 | 0.8995 | 0.8520 | 0.9160 |
| AdaLoRA | 0.9440 | 0.8637 | 0.9553 | 0.8952 | 0.9069 | 0.8736 | 0.9163 |
| SoRA | 0.9322 | 0.8095 | 0.9564 | 0.8540 | 0.8734 | 0.8777 | 0.9222 |
| PiSSA | 0.9443 | 0.8729 | 0.9621 | 0.9230 | 0.9150 | 0.8869 | 0.9200 |
| BOFT | 0.9423 | **0.9025** | **0.9644** | 0.9210 | 0.9016 | 0.8881 | 0.9192 |
| SVFT(P) | 0.9427 | 0.8969 | 0.9541 | 0.9016 | 0.8877 | 0.8724 | 0.9180 |
| SVFT(R) | 0.9390 | 0.8805 | 0.9002 | 0.9150 | 0.8899 | 0.8809 | 0.9173 |
| VeRA | 0.9324 | 0.8993 | 0.9553 | 0.9040 | 0.8794 | 0.8700 | 0.8871 |
| LoRA-XS | 0.8475 | 0.7802 | 0.9243 | 0.8038 | 0.8132 | 0.8065 | 0.8231 |
| LS-LoRA | 0.9235 | 0.8736 | 0.9427 | 0.8757 | 0.8382 | 0.6751 | 0.8767 |
| LoRETTA | 0.9325 | 0.8680 | 0.9553 | 0.8920 | 0.8873 | 0.7581 | 0.9066 |
| STAN (Ours) | **0.9477** | 0.8974 | 0.9622 | **0.9230** | **0.9166** | **0.9114** | **0.9277** |

Table 2: Results with DeBERTaV3-base across more baseline methods. The best results are highlighted in **bold**, and the second best results are underlined. All are measured by accuracy.

# 4 EXPERIMENTS

In this section, we conduct a series of experiments to validate the effectiveness of our proposed method, STAN. In Section 4.1, we evaluate its task performance across a range of benchmarks and model architectures, comparing it against state-of-the-art and representative PEFT methods to demonstrate its efficacy. In Section 4.2, we further explore the applicability of STAN to the Stable Diffusion 3 (SD3) model by fine-tuning it on a style alignment task, assessing its impact on both generation quality and adaptation flexibility. We also present qualitative visualizations that highlight the disentangled features learned by STAN, offering insight into its representational behavior. And we place more experiments related to interpretability in Appendix H. Finally, Section 4.3 presents ablation studies examining the role of sparsity levels and their influence on both performance and representation quality and in Section 4.4 we studies about the catastrophic forgetting phenomena comparing to LoRA. Appendix D analyzes the training dynamics and convergence behavior of STAN, focusing on its stability and efficiency during fine-tuning. All results are reproduced by us in a unified training environment using official implementations.

| Dataset | LLaVA-1.5-7B | | LLaVA-1.5-13B | | LLaVA-1.6-Vicuna-13B | |
|---|---|---|---|---|---|---|
| | LoRA | STAN (Ours) | LoRA | STAN (Ours) | LoRA | STAN (Ours) |
| **GQA** | 80.60 | **82.36** | 81.49 | **83.16** | 82.29 | **83.77** |
| **ScienceQA** | 89.17 | **90.25** | 91.34 | **92.96** | 92.06 | **92.49** |

Table 3: Performance comparison on multimodal (vision-language) tasks across three LLaVA model variants using the GQA and ScienceQA benchmarks. The best results are highlighted in **bold**.

| Method | GSM8K | MATH | Avg. |
|---|---|---|---|
| LoRA | 60.6 | 10.8 | 35.7 |
| PiSSA | 58.2 | 10.4 | 34.3 |
| STAN (Ours) | **60.7** | **11.6** | **36.15** |

Table 4: Performance on math & code benchmarks on LLaMA-2-7B. Best is in **bold**.

| Method | VizWiz | POPE | MMBench | Avg. |
|---|---|---|---|---|
| LoRA | 0.5021 | 0.8549 | 0.5403 | 0.6324 |
| STAN (Ours) | **0.5109** | **0.8611** | **0.5416** | **0.6379** |

Table 5: Performance on multimodal benchmarks on LLaVA-1.5-7B. The best results are highlighted in **bold**.

### 4.1 QUANTITATIVE COMPARISON

In this section, we compare our method against baseline approaches on single-modality (language), reasoning (math & code) and multi-modality (vision-language) benchmarks to demonstrate the validity and advantages of STAN. For language tasks, we compare STAN with LoRA (Hu et al., 2022), AdaLoRA (Zhang et al., 2023), and SoRA (Ding et al., 2023), using four pretrained large language models: RoBERTa-base/large (Liu et al., 2019) and DeBERTaV3-base/large (He et al., 2023b). These models are evaluated on five tasks from the GLUE benchmark (Wang et al., 2018). We report accuracy for MNLI, SST-2, QNLI, and QQP, and use Matthew's correlation for CoLA. Further, we present a more comprehensive comparison experiment with wider range of baseline methods (Hu et al., 2022; Zhang et al., 2023; Ding et al., 2023; Meng et al., 2024; Liu et al., 2023d; Lingam et al., 2024; Kopiczko et al., 2023; Bałazy et al., 2024; He et al., 2022; Yang et al., 2024) on DeBERTaV3-base (He et al., 2023b) with GLUE benchmark (Wang et al., 2018). The results are summarized in Table 1 and Table 2. For multimodal tasks, we evaluate STAN against LoRA on three variants of the LLaVA model (Liu et al., 2023a), using the GQA (Hudson & Manning, 2019) and ScienceQA (Lu et al., 2022) benchmarks. Moreover, we compare STAN against LoRA using more complex benchmarks, VizWiz (Gurari et al., 2018), POPE (Li et al., 2023) and MMBench (Liu et al., 2024b), on LLaVA-1.5-7b (Liu et al., 2023a), to further demonstrate the robustness of our method. All benchmarks are evaluated using accuracy and the results are summarized in Table 3 and Table 5. As for the reasoning task, we compare our STAN method with LoRA (Hu et al., 2022) and PiSSA (Meng et al., 2024) on GSM8K (Cobbe et al., 2021) and MATH (Hendrycks et al., 2021) benchmarks using LLaMA-2-7B (Touvron et al., 2023). The results are shown on Table 4. Experimental settings and runtime analysis are detailed in Appendix C.

As shown in the above Tables, the experimental results consistently demonstrate the superior performance and stability of our proposed method across a diverse range of tasks and model architectures. On language understanding benchmarks, STAN frequently achieves the highest scores across all five datasets when applied to different models of varying sizes and structures, often outperforming established PEFT methods by a noticeable margin. Notably, although SoRA (Ding et al., 2023) also incorporates sparsity, it exhibits significant instability during fine-tuning, whereas STAN delivers consistently strong performance (See more details in Appendix D). This pattern of performance extends to reasoning and vision-language tasks, where STAN consistently outperforms other methods across all benchmarks with different models. The consistent top-tier results across modalities, model types, model scales, and benchmarks highlight not only the enhanced performance of STAN, but also its robust stability and generalizability.

### 4.2 STYLE ALIGNMENT

In this section, we evaluate the capability of STAN to perform style alignment in the context of image generation. Our experiments are conducted on Stable Diffusion 3 (SD3) (Rombach et al., 2022), with a mixed dataset comprising images from WikiArt (Saleh & Elgammal, 2015) and the DualStyleGAN dataset (Yang et al., 2022), offering a diverse range of artistic styles.

We compare the performance of three approaches: the pretrained SD3 model without any fine-tuning (denoted as None), the SD3 model fine-tuned using LoRA, and the SD3 model fine-tuned using our proposed STAN method. To quantitatively evaluate style alignment, we employ two metrics:

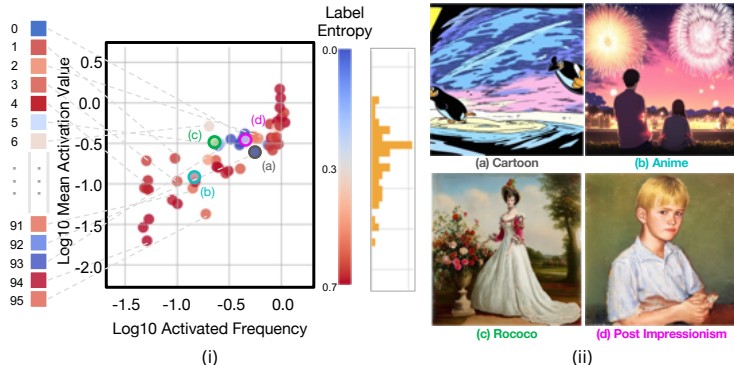

Figure 2: Qualitative demonstration of style alignment results across different methods using SD3. None denotes the pretrained model without fine-tuning.

Figure 3: Qualitative demonstration of the sparse representations learned by STAN. (i) Distribution of sparsely activated intermediate features across four distinct styles. (ii) Visualizations corresponding to each style: (a) Cartoon, (b) Anime, (c) Rococo, and (d) Post Impressionism.

CLIP-Score (Radford et al., 2021) and DINO-Score (Caron et al., 2021), both of which measure the semantic similarity between the generated image and the corresponding textual style prompt, but rely on different pretrained vision-language models. As shown in Table 6, STAN achieves the highest performance on both metrics, with a CLIP-Score of 0.6694 and a DINO-Score of 0.4283, outperforming both the baseline and the LoRA-fine-tuned variant.

To ensure that the observed quantitative improvements are also perceptually meaningful, we conducted an additional human user study focusing on both style alignment fidelity and overall visual quality in the SD3 experiments. We randomly sampled 20 distinct style prompts, and for each prompt, participants were presented with three images generated by the None, LoRA, and STAN methods with randomized orders. Users were asked to answer the question: Which image best matches the target style and has the highest overall visual quality?

The aggregated win rates, detailed in Table 7, strongly corroborate the preceding CLIP and DINO scores. STAN is preferred in over 90% of the comparisons, demonstrating a significant superiority in both style alignment and perceived image quality over the pretrained baseline and the LoRA-finetuned model. These results validate that the performance gains achieved by STAN are not merely numerical artifacts but translate to substantial enhancements in the human perception of the generated content. Combined with the disentanglement analysis in our style experiments, these findings further strengthen the claim that STAN learns disentangled, style-specific sparse features, supporting more precise and controllable generation.

| Methods | CLIP-Score↑ |
|---|---|
| None | 0.5572 |
| LoRA | 0.6509 |
| STAN (Ours) | **0.6694** |

| Methods | DINO-Score↑ |
|---|---|
| None | 0.3383 |
| LoRA | 0.4142 |
| STAN (Ours) | **0.4283** |

Table 6: Quantitative comparison of style alignment results using SD3. Best in **bold**.

| Method | Wins | Total | Win Rate (%) |
|---|---|---|---|
| None | 6 | 501 | 1.20 |
| LoRA | 39 | 501 | 7.78 |
| **Ours** | **456** | **501** | **91.02** |

Table 7: Human Evaluation of Style Alignment and Image Quality.

The superior quantitative results of STAN are further supported by visual evidence, as illustrated in Figure 2. The sparse latent space induced by STAN facilitates more effective disentanglement and encoding of stylistic elements, enabling more accurate alignment with the intended artistic styles. Visualizations show that the pretrained model often fails to capture or render the specified styles, while LoRA improves alignment but may still result in stylistic confusion. For example, blending characteristics of Realism and Impressionism, or generate inconsistent outputs in styles such as Cartoon or Illustration. In contrast, images produced by the STAN-fine-tuned model exhibit the most faithful and distinct adherence to the target styles, demonstrating its improved ability to isolate and apply diverse artistic features. These results highlight not only the enhanced representational capacity of STAN, but also its ability to decouple semantically similar yet stylistically distinct information. Additional visual examples are provided in Appendix E.

Furthermore, to highlight disentangled features learned by STAN, we present a statistical analysis in Figure 3. Part (i) illustrates activation frequency statistics of sparsely activated intermediate features when the model is prompted with four distinct artistic styles, i.e., Cartoon, Anime, Rococo, and Post Impressionism, using 300 generated samples per style. The y-axis represents $\log_{10}$ mean activation value, and the x-axis indicates $\log_{10}$ activation frequency. Each point is colored according to its label entropy, which reflects the degree of style specificity associated with that neuron: lower entropy values indicate specialization (i.e., the neuron is primarily activated by one style), while higher entropy suggests activation by a broader mix of styles. Details on the formulation of this analysis can be found in Appendix F. Part (ii) presents corresponding visual examples, illustrating that the images activating a given neuron are consistently aligned with its associated artistic style. The results reveal that different styles tend to activate distinct, often non-overlapping, subsets of sparse latent features. A small number of shared neurons appear to capture common generative priors, while the majority remain style-specific. This separation in feature activations suggests that STAN is effectively associating specific sparse components with semantically coherent stylistic concepts. Moreover, it demonstrates the expressive capacity of subspace combinations in the sparse latent space. To better demonstrate the interpretability of STAN, we did the similar experiments on the language task as well. Please refer to Appendix H for more details.

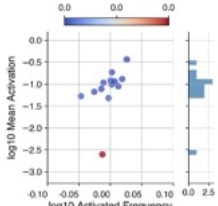

We also conduct additional experiments where we apply LoRA under the same setup and probe its latent directions using the same attribution and activation-frequency analysis. As demonstrated in Figure 4, The dense low-rank adaptations lead to more entangled features compared to STAN's sparse latent units, and we can not separate styles according to the middle features like we did to STAN.

The ability of STAN to map these diverse styles to separable sparse feature activations is a strong indication of its capacity for learning disentangled features. Compared to LoRA, this broader representational capacity is essential for achieving fine-grained control in generative modeling.

Figure 4: Interpretablity experiment using LoRA.

## 4.3 ABLATION STUDY

In this section, we conduct the ablations to investigate the impact of the Top$K$ parameter along with the middle latent dimension $l$ on the performance of our STAN method. The evaluation is based on accuracy on a subset of the MNLI dataset (Wang et al., 2018). We control the size of the latent space using an expansion factor, which scales the dimensionality of the intermediate representation by multiplying it with the input feature dimension. The Top$K$ parameter controls the sparsity level by selecting the $K$ most active features in this latent space. In our experiments, we vary the expansion factor across $1, 1.5, 2$ and Top$K$ across $1, 4, 8, 16, 32, 64$, along with a setting denoted as $n$-latents, where all intermediate features are retained (i.e., no sparsification is applied). The full results of the ablation study are presented in Table 8.

As in Table 8, we can see a nuanced interplay between these two hyperparameters. For all three expansion factors, accuracy peaks at $K$=8 or $K$=16, which indicates that as the latent space capacity increases, a slightly larger number of active features might be beneficial, but it does not mean the elevation of active feature size will bring the better performance.

| TopK | Expansion Factor | | |
|---|---|---|---|
| | 1 | 1.5 | 2 |
| 1 | 0.8139 | 0.8088 | 0.8131 |
| 4 | 0.8158 | 0.8122 | 0.8149 |
| 8 | **0.8192** | 0.8206 | 0.8177 |
| 16 | 0.8171 | 0.8180 | 0.8213 |
| 32 | 0.8173 | 0.8170 | 0.8169 |
| 64 | 0.8162 | 0.8155 | 0.8161 |
| $n$-latents | 0.8158 | **0.8262** | **0.8286** |

Table 8: Ablation study on expansion factor and Top$K$. Best is in **bold**, and second is in underlined.

Another finding emerges when comparing the sparse Top$K$ configurations with the $n$-latents baseline. Even when the it configuration utilizes a substantially larger, high-rank intermediate dimension, with expansion factors equal to 1.5 and 2, achieving accuracies of 0.8262 and 0.8286 respectively, STAN configurations with a small Top$K$ value can achieve remarkably comparable results. This implies that a significant portion of the features within the larger, unsparsified latent space may be redundant for the given task. The superior performance of STAN with small $K$ values, even when the potential latent dimension is large, underscores the efficacy of sparsity, it allows the model to achieve competitive results by identifying a compact set of the most salient features, thereby alleviating the need for an excessively large number of active parameters and demonstrating that sparsity is a sensible and efficient approach to parameter utilization.

### 4.4 CATASTROPHIC FORGETTING

To directly evaluate the catastrophic forgetting, we conducted additional bidirectional sequential finetuning experiments on two tasks: SST-2 and MNLI. We perform sequential finetuning in two directions: SST-2 $\rightarrow$ MNLI: Finetune using STAN on SST-2 first, then continue training on MNLI while monitoring SST-2 accuracy. MNLI $\rightarrow$ SST-2: Finetune on MNLI, then finetune on SST-2 while monitoring MNLI accuracy. This setup directly measures forgetting of the first task during learning of the second. Table 9 demonstrates the experiment results.

We can see that for SST-2 $\rightarrow$ MNLI, only a 4.6-point drop after 20 epochs, and for MNLI $\rightarrow$ SST-2, it's a moderate 6.5-point decline. The retults indicate a very mild forgetting. We also performed LoRA based experiments utlizing the same bidirection setting, its drop for SST-2 $\rightarrow$ MNLI is 14.2 points after 20 epochs and for MNLI $\rightarrow$ SST-2 the drop is 17.8 points. That is because STAN's architecture naturally mitigates forgetting due to (1) sparse latent activations that localize task-specific updates, (2) minimal interference across tasks, since only a small subset of neurons is modified. (3) disentangled and interpretable features, allowing different tasks to occupy different sparse subspaces. This is fundamentally different from LoRA's dense low-rank updates, which modify shared directions and are more prone to overwriting previous knowledge.

| method | Accuracy (%) at Epoch | | | | |
|---|---|---|---|---|---|
| | 1 | 5 | 10 | 15 | 20 |
| SST-2 | 85.00 | 84.03 | 82.83 | 81.62 | 80.41 |
| MNLI | 80.00 | 78.62 | 76.90 | 75.17 | 73.45 |

Table 9: Catastrophic forgetting study using STAN. The first row is the accuracy change of SST-2 in SST-2 $\rightarrow$ MNLI setting the the second row is the accuracy change of MNLI in MNLI $\rightarrow$ SST-2.

## 5 CONCLUSION AND FUTURE WORKS

In this paper, we introduced STAN, a novel parameter-efficient fine-tuning method designed to support dynamic feature selection and address key limitations of existing methods – particularly the restricted representational capacity inherent in low-rank adaptation paradigms such as LoRA. By leveraging a sparse autoencoder architecture, STAN dynamically encodes task-specific adaptations through a high-dimensional yet sparse set of features. This enables a more flexible and expressive adaptation mechanism compared to rigid low-rank projections. For future work, STAN can be extended to a broader range of tasks and modalities, including more complex reasoning and multi-step decision-making settings. In addition, further direction can be explored, such as investigating alternative sparsity-inducing mechanisms beyond the current Top$K$ selection strategy. For the use of large language models, please refer to Appendix I.

**Ethics statement.** We have adhered to the ICLR Code of Ethics. Our research primarily utilizes publicly available datasets and pretrained models, and we do not foresee any direct negative societal impacts or ethical concerns arising from our methodology.

**Reproducibility statement.** We are committed to ensuring the full reproducibility of our research. To facilitate this, the complete source code to replicate experiments presented in this paper will be made publicly available upon publication.

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

# Appendix to
# The Chosen Few: Sparse Adaptation for Fine-Tuning Large Models

## A  SUFFICIENT CONDITIONS FOR STAN INTERPRETABILITY AND IDENTIFIABILITY

**Notation.**  Let $E \in \mathbb{R}^{l \times n}$ and $D \in \mathbb{R}^{m \times l}$. For any input $x \in \mathbb{R}^n$, define

$$z(x) := Ex, \qquad I(x) := T_k\big(|z(x)|\big), \qquad S(x)_{ii} = \mathbb{1}\{i \in I(x)\}. \tag{7}$$

The STAN adaptation operator and output are

$$\mathcal{A}(x) := D\,S(x)\,E, \qquad \Delta W\,x = \mathcal{A}(x)\,x. \tag{8}$$

For a standard basis vector $e_p$ with $p \in [n]$, write $|(Ee_p)_i|$ for the singleton response at latent row $i$.

**Assumptions used below. They are sufficient, not necessary.**

1. Singleton margin:

$$i^\star(p) := \arg\max_{i \in [l]} |(Ee_p)_i|, \qquad \gamma_p := |(Ee_p)_{i^\star(p)}| - \max_{j \neq i^\star(p)} |(Ee_p)_j| \;>\; 0 \quad \text{for all } p \in [n]. \tag{9}$$

2. Blockwise identity on singletons:

$$D_{\{i^\star(p)\}}\,E_{\{i^\star(p)\}}\,e_p \;=\; e_p \quad \text{for all } p \in [n]. \tag{10}$$

3. Row orthonormality for identifiability:

$$EE^\top \;=\; I_l. \tag{11}$$

**Theorem 1. Singleton exact recovery implies monosemanticity.** Under Assumptions 1 and 2, for every $p \in [n]$ and every $\alpha > 0$,

$$i^\star(p) \in I(\alpha e_p), \tag{12}$$

$$\Delta W(\alpha e_p) = D\, S(\alpha e_p)\, E(\alpha e_p) = D_{\{i^\star(p)\}}\, E_{\{i^\star(p)\}}(\alpha e_p) = \alpha e_p. \tag{13}$$

Moreover, for any $x$ with $p \notin \mathrm{supp}(x)$, one has $i^\star(p) \notin I(x)$, hence latent $i^\star(p)$ is monosemantic for coordinate $p$.

*Proof. One line.* Since $\gamma_p > 0$, Top-$k$ on $|E(\alpha e_p)|$ selects $i^\star(p)$; substituting the blockwise identity yields the claim.

**Theorem 2. Identifiability up to permutation and positive scaling.** Assume Assumptions 1 and 2 for all $p \in [n]$, and also assume Assumption 3. If another parameter pair $(\tilde{D}, \tilde{E})$ satisfies the same singleton conditions, then there exist a permutation matrix $P$ and a positive diagonal matrix $\Lambda$ such that

$$\tilde{E} = \Lambda\, P\, E, \qquad \tilde{D} = D\, P^\top \Lambda^{-1}. \tag{14}$$

In particular, the set of singleton carriers $\{i^\star(p) : p \in [n]\}$ is unique up to permutation and positive rescaling, which means STAN is identifiable modulo permutation and scale.

**Remark.** The three assumptions above describe a sufficient regime for interpretability and identifiability. They can be checked post hoc by inspecting singleton margins, testing the singleton blockwise identity, and measuring $\|EE^\top - I_l\|_F$.

## B MORE DISCUSSION ON RELATED METHODS

In this section, we differentiate our proposed STAN from other proposed PEFT methods, including AdaLoRA (Zhang et al., 2023), and SoRA (Ding et al., 2023).

- **AdaLoRA** (Zhang et al., 2023):

  AdaLoRA parameterizes the weight update in an SVD-like form $\Delta W = P\Lambda Q$, where $\Lambda$ contains trainable singular values that are adaptively pruned based on sensitivity scores. This allows the model to adjust the effective rank of the update matrix during training:

  $$W = W^{(0)} + \Delta W = W^{(0)} + P\Lambda Q, \tag{15}$$

  In contrast, our STAN operates in a high-dimensional latent space and performs adaptation by *dynamically* selecting a sparse subset of features for each input using the non-linear topk operator, rather than adjusting the overall rank during training. While AdaLoRA seeks a low-rank approximation constrained by a parameter budget, STAN focuses on sparse, input-dependent feature selection. As a result, STAN enables a more flexible and adaptive mechanism, whereas AdaLoRA offers a more static and structured approach.

- **SoRA** (Ding et al., 2023):

  SoRA extends the standard LoRA decomposition by introducing a learnable sparse gating vector $g \in \mathbb{R}^{r_{\max}}$ between the down- and up-projection matrices. This gating mechanism allows the model to adaptively select its effective rank during training. Given an input feature $x \in \mathbb{R}^q$, the forward pass of a single SoRA block is defined as:

  $$z = W_u\big(g \odot (W_d x)\big), \tag{16}$$

  where $W_d \in \mathbb{R}^{r_{\max} \times q}$ and $W_u \in \mathbb{R}^{p \times r_{\max}}$ are the projection matrices and $\odot$ denotes element-wise multiplication. The gating is regularized toward sparsity through a proximal-gradient soft-thresholding update:

  $$g^{t+1} = \mathcal{T}_{\eta_t \lambda}\big(g^t - \eta_t \nabla_g \mathcal{L}_0(\Delta^t)\big), \qquad \mathcal{T}_\xi(x) = \begin{cases} x - \xi, & x > \xi, \\ 0, & |x| \le \xi, \\ x + \xi, & x < -\xi, \end{cases} \tag{17}$$

  where $\mathcal{L}_0$ is the original task loss, $\eta_t$ is the learning rate, and $\lambda$ controls sparsity.

SoRA performs static, global pruning of entire rank components. In contrast, STAN adopts a dynamic, input-dependent feature selection mechanism using the $\mathrm{topk}$ operator, activating different $k$-dimensional subspaces for different inputs without permanently pruning features during training. This design offers greater flexibility while maintaining training stability, as evidenced by the results shown in Figure 7.

- **LS-LoRA** (He et al., 2022):

  While both methods leverage sparsity, they are fundamentally different in their approach and philosophy. LS-LoRA employs a static pruning strategy, using an algorithm like SNIP to remove a fixed set of weights once at initialization. Consequently, its sparse pattern is predetermined and remains unchanged throughout the entire training process. In contrast, STAN operates on the principle of dynamic feature selection. Its sparsity is not static but is instead flexible and input-dependent, emerging from the data representations during training. This core difference reflects their distinct goals: LS-LoRA focuses on effective compression, whereas STAN is designed for conceptual decomposition and representation selection, allowing the model to dynamically activate features relevant to a given input.

## C  Experimental Setting and Runtime Analysis

In this section, we perform a runtime analysis and detail the experimental settings used throughout our evaluations, spanning three task categories: (1) natural language understanding, (2) vision-language reasoning, and (3) text-to-image generation.

### C.1  Runtime Analysis

To provide a comprehensive evaluation of STAN's practical efficiency, we analyze its performance in terms of both training throughput and inference latency.

First, we compare STAN against other adaptive sparse fine-tuning methods, namely AdaLoRA and SoRA, under the same experimental settings: 3 epochs toy training for SST-2 dataset with 1 A100 GPU with batch size equals to 32 and sequence length equals to 256 on RoBERTa-base. The results, presented in Table 10, demonstrate that STAN achieves superior training throughput compared to both baselines. For instance, it processes nearly 50% more tokens per second than AdaLoRA. Crucially, this training acceleration is achieved while maintaining an inference latency that is on par with these methods, showcasing its superior training efficiency without compromising inference speed.

| Method | Inference Latency (ms) | Train Tokens/sec | GPU-hours (3 ep) |
|---|---|---|---|
| STAN | 23.16 | 19815 | 0.01 |
| AdaLoRA | 23.29 | 13721 | 0.01 |
| SoRA | 25.17 | 14451 | 0.015 |

Table 10: Efficiency comparison with adaptive sparse methods AdaLoRA and SoRA. STAN shows significantly higher training throughput while maintaining comparable inference latency.

| Method | Avg. Inference Latency/sample (s) | Train Tokens/sec | GPU-hours (3 ep) | Peak GPU Memory (GB) |
|---|---|---|---|---|
| STAN | 0.419 | 2359.42 | 0.01 | 39.62 |
| LoRA | 0.404 | 1533.81 | 0.02 | 33.53 |

Table 11: Efficiency comparison with LoRA on the LLaVA-1.5-13B model. STAN demonstrates superior training efficiency with only a minor trade-off in peak memory.

To further assess the scalability and practical benefits of our approach, we conduct an additional experiment on a larger model, LLaVA-1.5-13B, comparing STAN directly with the widely-used LoRA. As detailed in Table 11, STAN maintains a significant advantage in training efficiency over LoRA even on this larger scale. Specifically, STAN achieves a higher training throughput and requires

approximately half the GPU-hours to complete the same training task. While inference latency is comparable to LoRA, STAN incurs only a modest increase in peak GPU memory usage.

To provide a rigorous evaluation of the trade-off between computational cost and task performance, we visualize the training trajectories of STAN against baselines.

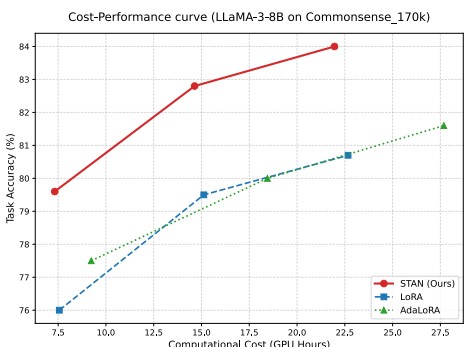

Figure 5: Cost-Performance curve on LLaMA-3-8B.

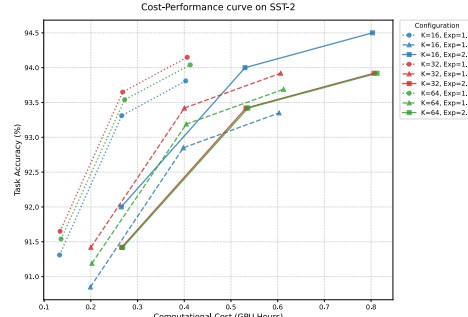

Figure 6: Robustness analysis across parameter budgets on SST-2.

As illustrated in Figure 5, on the large-scale LLaMA-3-8B benchmark, STAN exhibits superior convergence dynamics. It achieves a significantly higher peak accuracy of 84.0% while requiring less total wall-clock training time (21.96 GPU hours) compared to LoRA (80.7% accuracy in 22.67 hours) and AdaLoRA. This empirical evidence highlights that the sparse high-dimensional features in STAN facilitate faster learning of task-relevant concepts, effectively offsetting the storage overhead.

Furthermore, Figure 6 investigates the method's robustness under varying parameter budgets. We observe distinct trends in runtime cost: increasing the Expansion Factor (which scales total trainable parameters) leads to a proportional increase in GPU hours (shifting curves to the right), while increasing $K$ introduces a minor additional sorting overhead. Despite these variations in computational cost, STAN maintains consistently high performance across different configurations. Notably, efficient settings with lower Expansion Factors (e.g., 1.0) yield optimal accuracy with minimal runtime, confirming that the performance gains stem from the sparse adaptation mechanism itself rather than mere parameter scaling.

## C.2 Natural Language Understanding

**Models and Benchmarks.** We evaluate on five representative tasks from the GLUE benchmark (Wang et al., 2018): MNLI, QNLI, SST-2, CoLA, and QQP. For each task, we follow the official train/test splits. Our experiments are conducted using four pretrained language models: RoBERTa-base, RoBERTa-large(Liu et al., 2019), DeBERTaV3-base, and DeBERTaV3-large(He et al., 2023b).

We attribute this compelling performance profile to our design. On large-scale models, the benefits of our dynamic sparsity for accelerating training become more pronounced. During inference, the substantial computational overhead of the base model itself diminishes the relative impact of the small overhead introduced by STAN's mechanism. These results underscore STAN's scalability and its practical advantages in large-model scenarios, making it a strong choice for efficient fine-tuning and deployment.

**Implementation Details.** During fine-tuning, we apply a weight decay of 0.1 and set the warmup ratio to 0.06. Optimization is performed using the AdamW optimizer with a linear learning rate scheduler. All experiments are conducted with a fixed random seed of 0 for reproducibility. The detailed training hyperparameters for all methods are summarized in Table 12.

| Method | Model | Epochs | Batch Size | LR | $r$ / $k$ | $\alpha$ | Lambda | Lambda2 |
|--------|-------|--------|------------|-----|-----------|----------|--------|---------|
| AdaLoRA | RoBERTa-base | 15 | 32 | 5e-4 | 2 | 32 | – | – |
| | RoBERTa-large | 15 | 32 | 5e-4 | 2 | 32 | – | – |
| | DeBERTaV3-base | 15 | 32 | 5e-4 | 2 | 32 | – | – |
| | DeBERTaV3-large | 15 | 32 | 5e-4 | 2 | 32 | – | – |
| SORA | RoBERTa-base | 15 | 32 | 8e-4 | 8 | 16 | 10 | 3e-4 |
| | RoBERTa-large | 15 | 32 | 8e-4 | 8 | 16 | 10 | 3e-4 |
| | DeBERTaV3-base | 15 | 32 | 8e-4 | 8 | 16 | 10 | 3e-4 |
| | DeBERTaV3-large | 15 | 32 | 8e-4 | 8 | 16 | 10 | 3e-4 |
| LoRA | RoBERTa-base | 15 | 32 | 5e-4 | 8 | 16 | – | – |
| | RoBERTa-large | 15 | 32 | 5e-4 | 8 | 16 | – | – |
| | DeBERTaV3-base | 15 | 32 | 5e-4 | 8 | 16 | – | – |
| | DeBERTaV3-large | 15 | 32 | 5e-4 | 8 | 16 | – | – |
| STAN (Ours) | RoBERTa-base | 15 | 32 | 5e-4 | 8 | – | – | – |
| | RoBERTa-large | 15 | 32 | 5e-4 | 8 | – | – | – |
| | DeBERTaV3-base | 15 | 32 | 5e-4 | 8 | – | – | – |
| | DeBERTaV3-large | 15 | 32 | 5e-4 | 8 | – | – | – |

Table 12: Summary of training hyperparameters used for each method and backbone model. Parameters not applicable to a given method are indicated with -.

### C.3 VISION-LANGUAGE REASONING

**Models and Benchmarks.** We conduct our experiments using three versions of the LLaVA (Liu et al., 2023a) model: LLaVA_1.5_7B, LLaVA_1.5_13B, and LLaVA_1.6_Vicuna_13B. We fine-tune these models on two benchmarks, GQA (Hudson & Manning, 2019) and ScienceQA (Lu et al., 2022), for each task we follow the official train/test splits provided.

**Implementation Details.** For fine-tuning the LLaVA models, we use the LLaVA_1.5_13B checkpoint equipped with the CLIP-ViT-L/14-336 vision encoder and an MLP-based projector. Training is conducted using DeepSpeed ZeRO-3 for efficient large-scale optimization. Detailed training hyperparameters for all LLaVA models and methods are provided in Table 13.

| Method | Model | Epochs | Batch Size | LR | $r$ / $k$ | $\alpha$ | Num Latents | Warmup Ratio |
|--------|-------|--------|------------|-----|-----------|----------|-------------|--------------|
| LoRA | LLaVA_1.5_7B | 15 | 16 | 2e-4 | 128 | 256 | – | 0.03 |
| | LLaVA_1.5_13B | 15 | 16 | 2e-4 | 128 | 256 | – | 0.03 |
| | LLaVA_1.6_Vicuna-13B | 15 | 16 | 2e-4 | 128 | 256 | – | 0.03 |
| STAN (Ours) | LLaVA_1.5_7B | 15 | 16 | 2e-4 | 128 | – | 512 | 0.03 |
| | LLaVA_1.5_13B | 15 | 16 | 2e-4 | 128 | – | 512 | 0.03 |
| | LLaVA_1.6_Vicuna-13B | 15 | 16 | 2e-4 | 128 | – | 512 | 0.03 |

Table 13: Summary of training hyperparameters for each method and backbone model. Parameters not applicable to a given method are indicated with -.

### C.4 TEXT-TO-IMAGE GENERATION

**Models and Benchmarks.** Experiments are conducted using the publicly available Stable Diffusion 3 (SD3) (Rombach et al., 2022) released by Stability AI, trained on a composite dataset comprising images from WikiArt(Saleh & Elgammal, 2015) and the DualStyleGAN dataset(Yang et al., 2022). The WikiArt dataset contains a broad collection of real-world artworks spanning diverse styles and historical periods, while DualStyleGAN provides synthetically generated images with fine-grained, diverse, controllable stylistic attributes.

**Implementation Details.** The training hyperparameters corresponding to all diffusion models and methods are summarized in Table 14.

| Method | Model | Resolution | Steps | Batch Size | Grad Accum. | LR | $r$ / $k$ | $\alpha$ | Num Latents |
|--------|-------|-----------|-------|-----------|-------------|-----|-----|-----|-------------|
| LoRA | SD3 | 1024 | 3000 | 1 | 1 | 4e-4 | 32 | 1 | – |
| STAN (Ours) | SD3 | 1024 | 3000 | 1 | 1 | 4e-4 | 96 | – | 512 |

Table 14: Summary of training hyperparameters for each method and backbone model. Parameters not applicable to a given method are indicated with -.

## D  ADDITIONAL TRAINING LOSS COMPARISONS

Beyond performance, the stability of the fine-tuning process is a critical indicator of a PEFT method's robustness and reliability. As previously highlighted in Table 1, although methods such as SoRA (Ding et al., 2023) incorporate sparsity, their practical utility is often limited by unstable training dynamics.

In this section, we present additional loss curves for comparison. Across all demonstrations, STAN consistently exhibits stable convergence, whereas other methods display either significant oscillations or fail to converge effectively. Notably, in some cases, the loss for SoRA and LoRA approaches zero, suggesting potential overfitting during the fine-tuning process. This indicates a limitation in representational capacity, highlighting the challenges of capturing task-relevant features under constrained adaptation schemes.

To further investigate this, The following Figures present training loss curves on the different datasets for STAN, compared against three other PEFT methods across different base models. Across all model architectures, STAN exhibits remarkably stable and smooth convergence. The training loss decreases consistently, without significant oscillations, and generally reaches or surpasses the final loss values achieved by LoRA and AdaLoRA. In contrast, the loss curves for SoRA display notable volatility and often plateau at higher values or even diverge – corroborating earlier observations of its instability. This erratic behavior during training likely contributes to SoRA's inconsistent and sometimes suboptimal task performance. By comparison, the stable convergence of STAN not only strengthens confidence in its training dynamics but also underpins the consistent, high performance reported in prior sections. These findings suggest that STAN's approach to sparse adaptation is inherently more robust, making it a more reliable and effective choice for parameter-efficient fine-tuning.

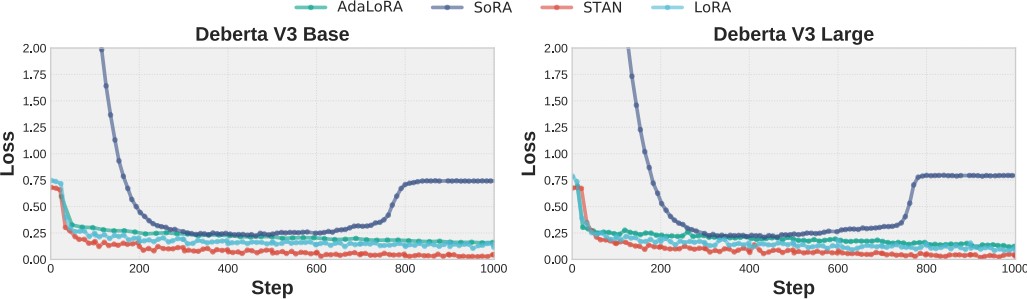

Figure 7: Training loss comparison on SST-2 using DeBERTaV3-base/large (He et al., 2023b). STAN consistently achieves the lowest loss, demonstrating greater stability compared to other methods.

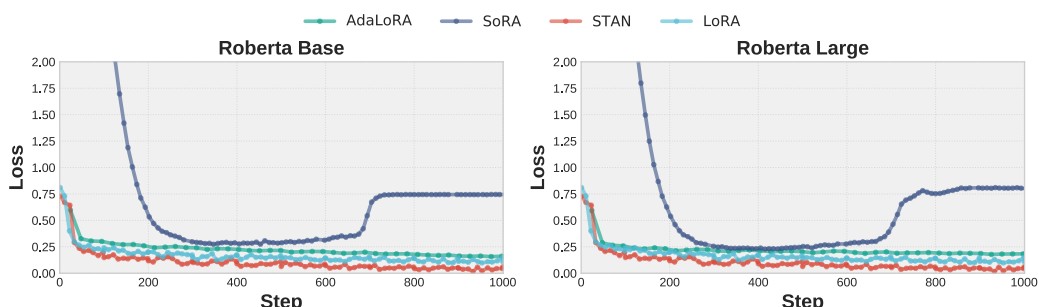

Figure 8: Loss comparison on SST-2 using RoBERTa-base and RoBERTa-large (Liu et al., 2019). STAN exhibits the lowest and most stable loss curve, while LoRA and AdaLoRA show smooth but slightly higher loss trajectories. In contrast, SoRA demonstrates instability and fails to converge effectively.

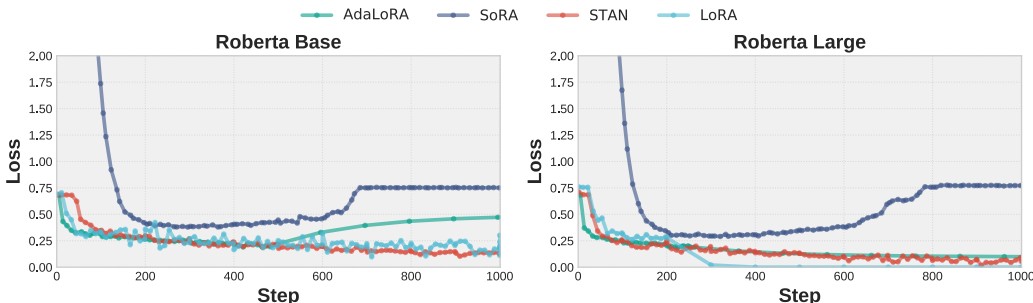

Figure 9: Loss comparison on QNLI using RoBERTa-base and RoBERTa-large (Liu et al., 2019). STAN achieves the lowest and most stable loss curve. LoRA exhibits more oscillations and signs of overfitting toward the end of training. In contrast, both AdaLoRA and SoRA show unstable convergence and fail to reach optimal performance.

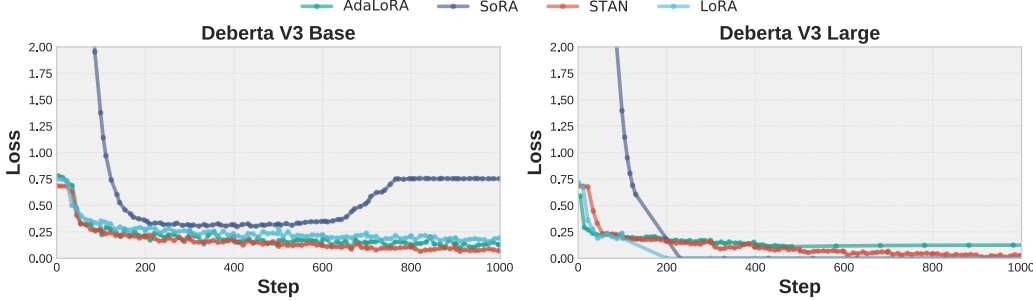

Figure 10: Loss comparison on QNLI using DeBERTaV3-base and DeBERTaV3-large (He et al., 2023b). STAN achieves the lowest and most stable loss trajectory. LoRA exhibits greater oscillation during training, while AdaLoRA shows signs of overfitting early on. SoRA remains unstable throughout and also displays indications of overfitting.

# E    ADDITIONAL QUALITATIVE RESULTS FOR DIFFUSION MODELS

In this section, we present additional images generated by SD3, fine-tuned using different methods and conditioned on different prompts.

STYLE : abstract expressionism
CONTENT : ladies with watermelons

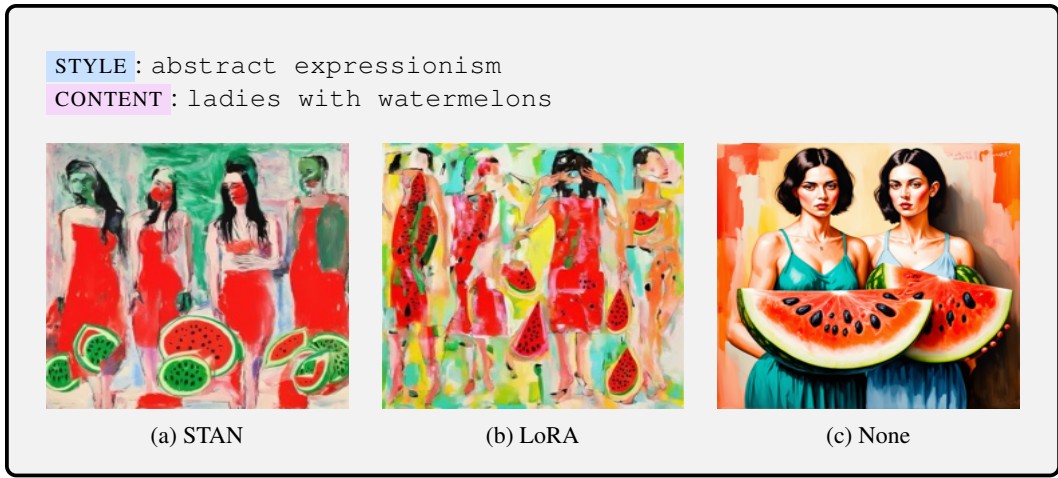

(a) STAN                (b) LoRA                (c) None

STYLE : anime
CONTENT : two students walking home after school in spring

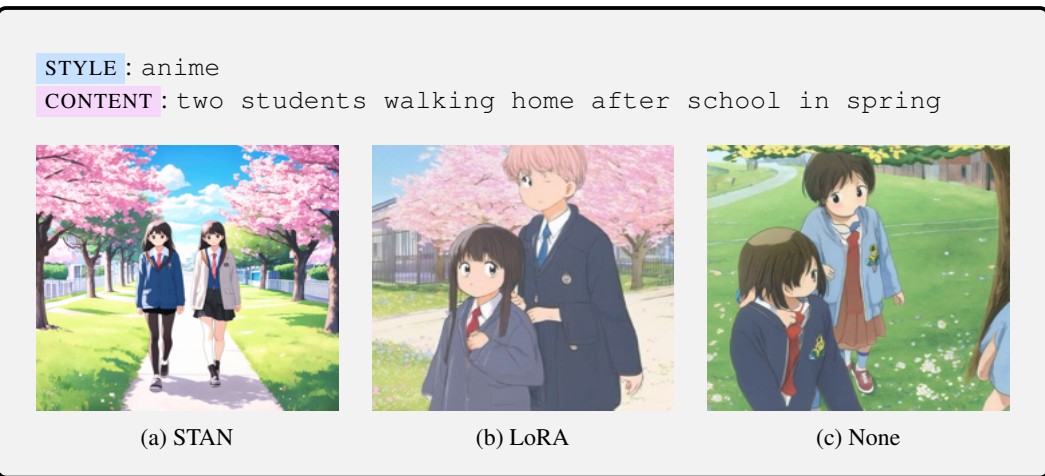

(a) STAN                (b) LoRA                (c) None

STYLE : baroque
CONTENT : women in robes

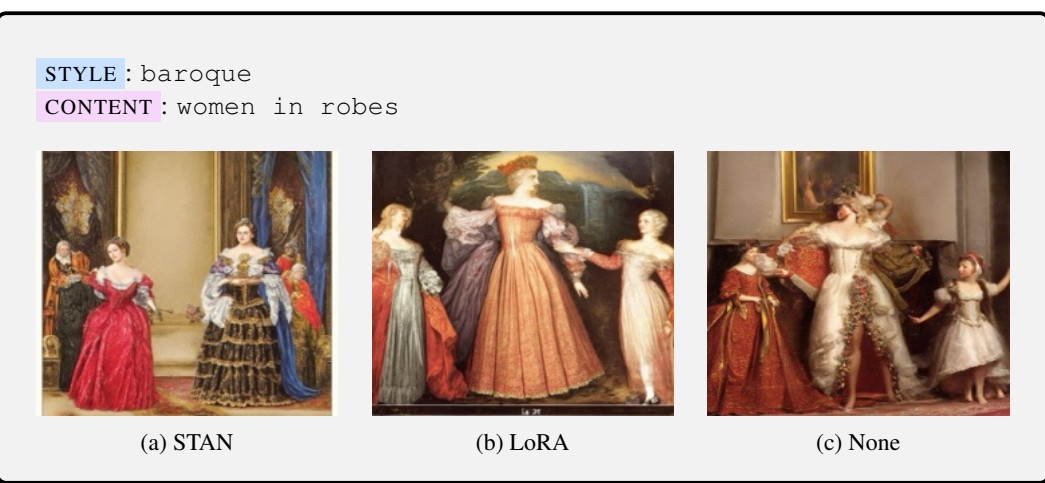

(a) STAN     (b) LoRA     (c) None

STYLE : cartoon
CONTENT : two kids building a time machine in a garage

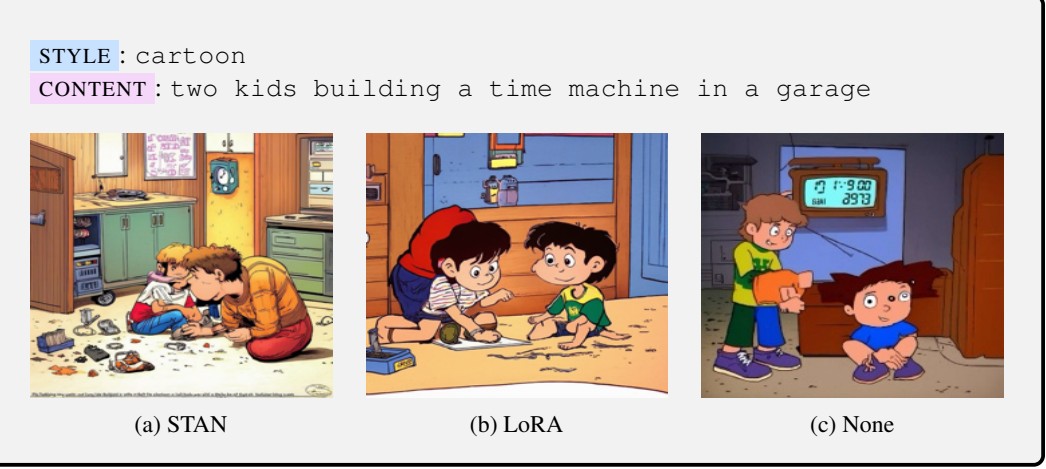

(a) STAN     (b) LoRA     (c) None

STYLE : ecole de paris
CONTENT : portrait

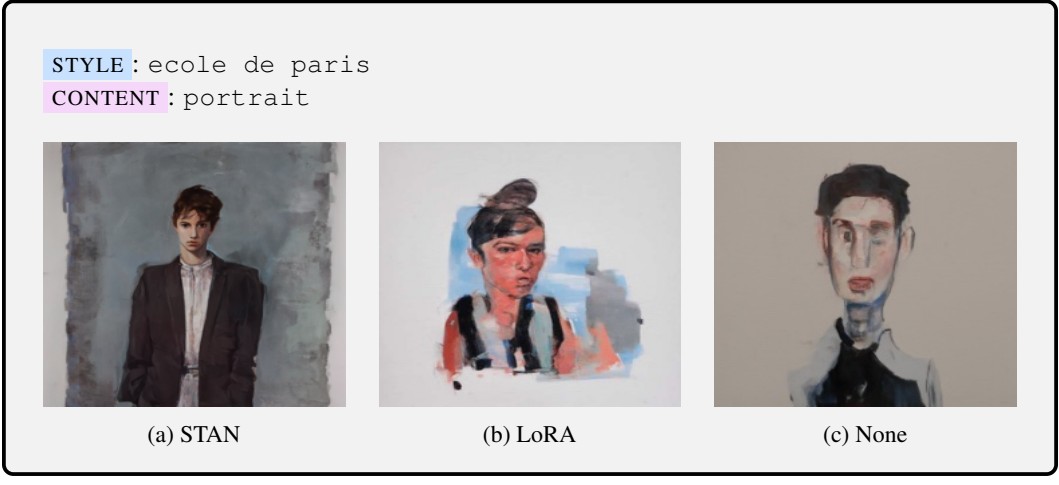

(a) STAN     (b) LoRA     (c) None

STYLE : expressionism
CONTENT : snowy ground

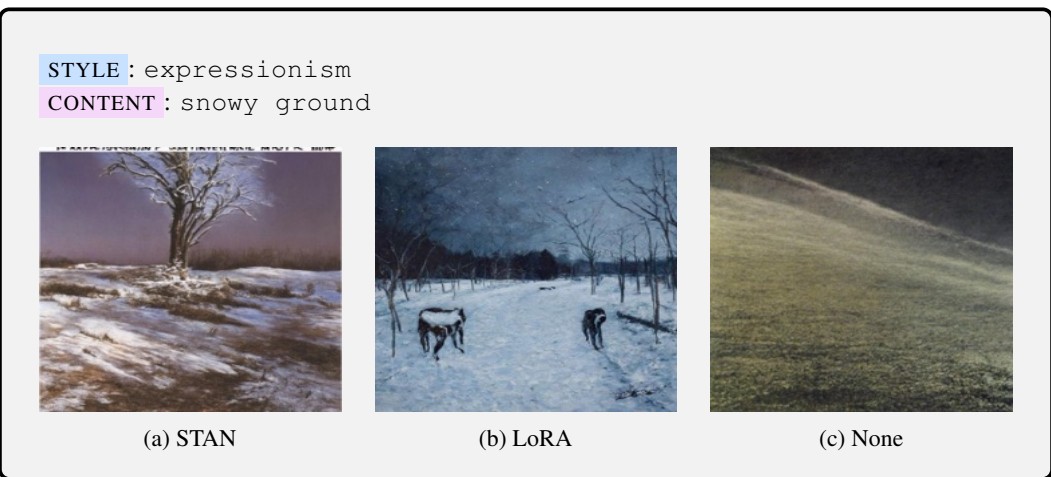

(a) STAN          (b) LoRA          (c) None

STYLE : fantasy
CONTENT : a castle carved into a cliff glowing with runes

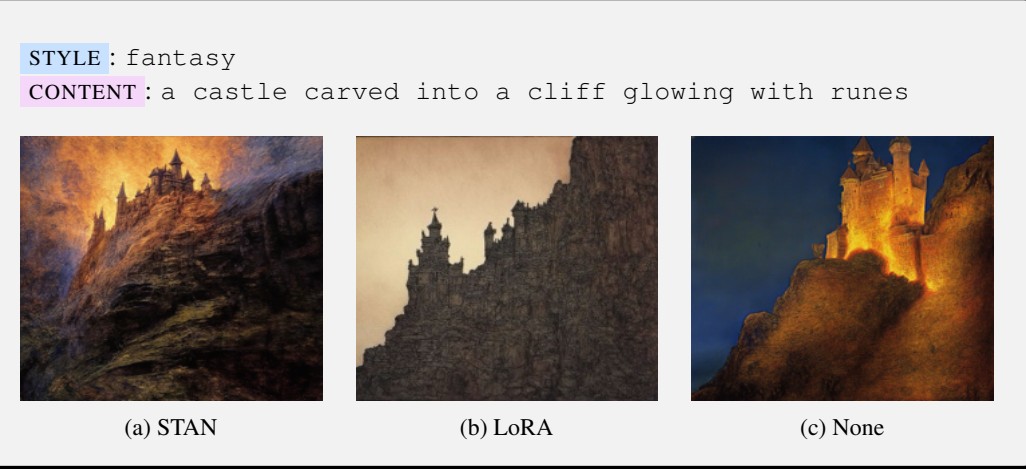

(a) STAN          (b) LoRA          (c) None

STYLE : illustration
CONTENT : a rainy day with two friends playing chess

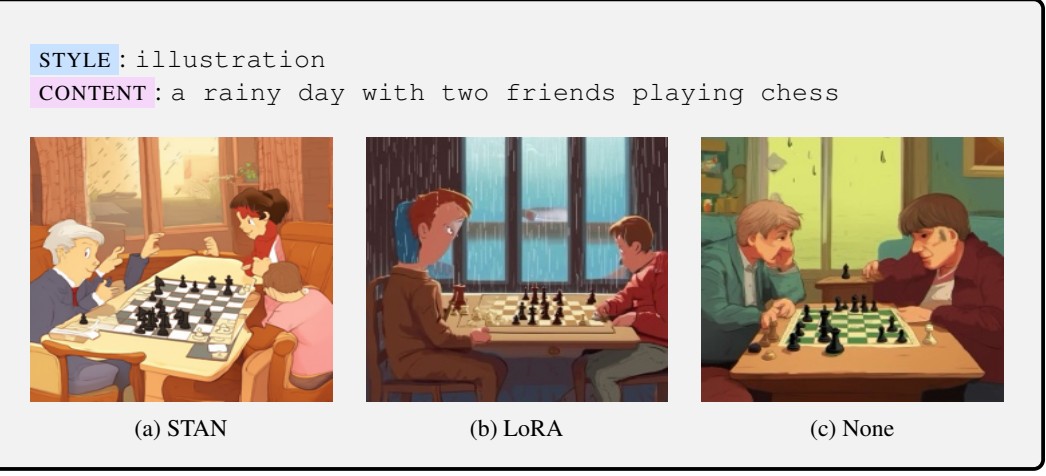

(a) STAN          (b) LoRA          (c) None

STYLE : impasto
CONTENT : a village with twisted trees and stars overhead

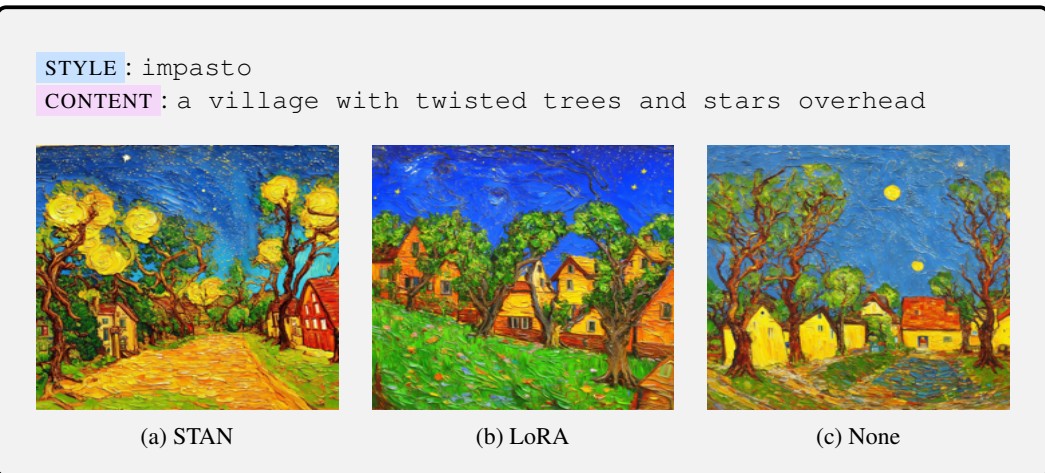

(a) STAN                          (b) LoRA                          (c) None

STYLE : impressionism
CONTENT : woman in the gardon

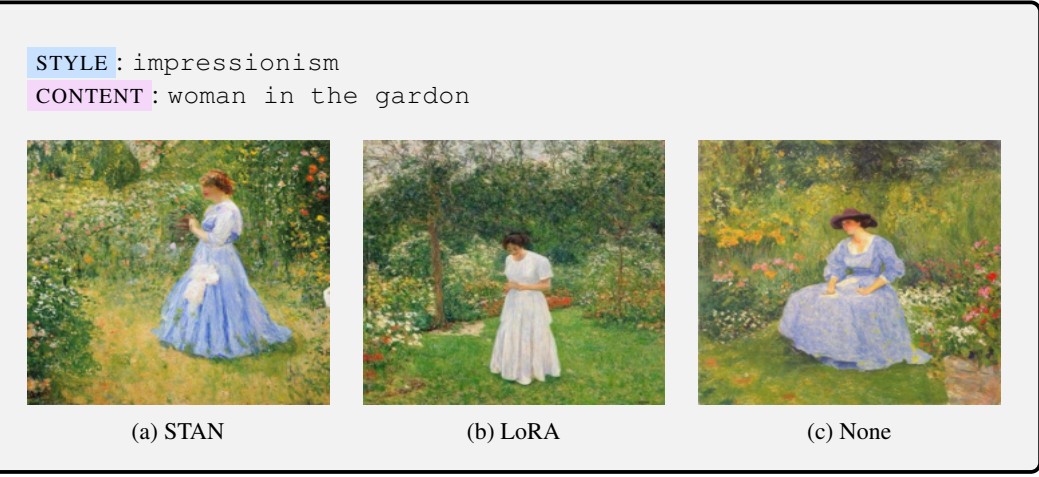

(a) STAN                          (b) LoRA                          (c) None

STYLE : naive art primitivism
CONTENT : blue eyed flowers

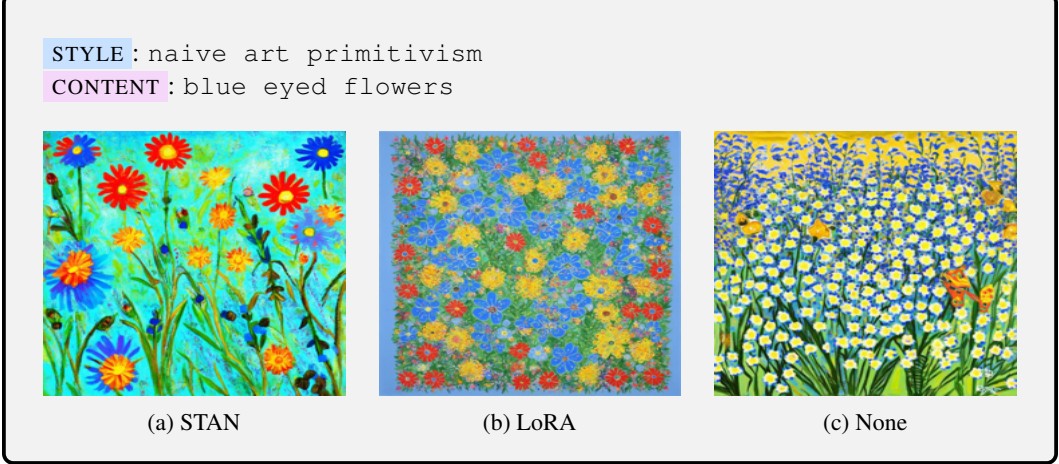

(a) STAN                          (b) LoRA                          (c) None

STYLE : neo impressionism
CONTENT : a women

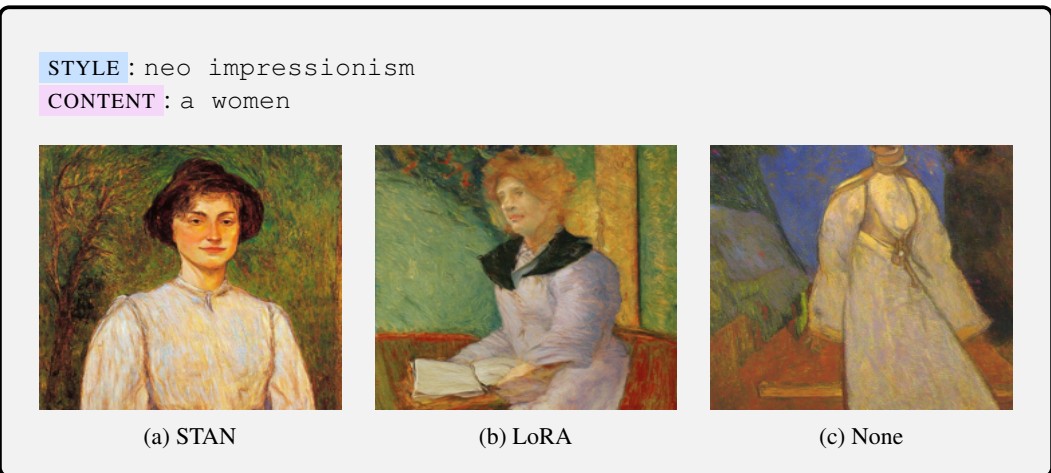

(a) STAN          (b) LoRA          (c) None

STYLE : neoclassicism
CONTENT : two women

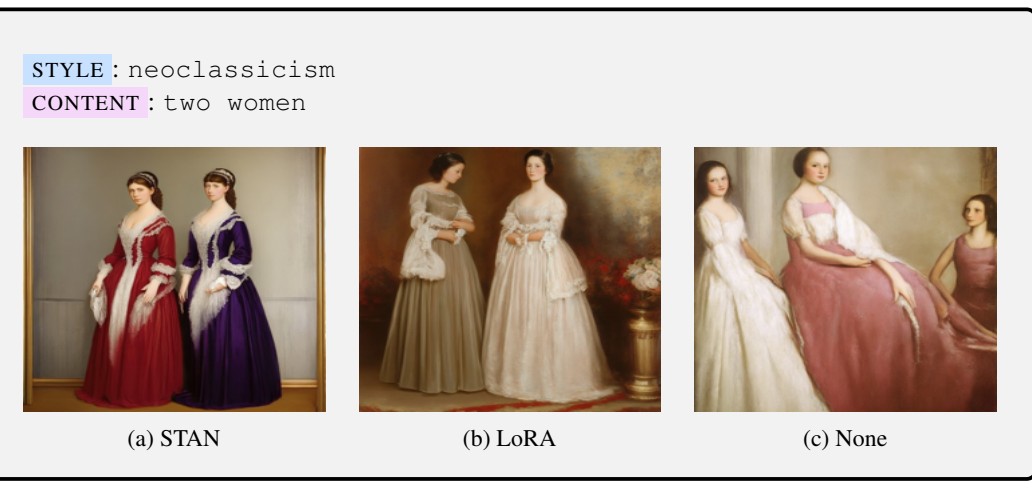

(a) STAN          (b) LoRA          (c) None

STYLE : post impressionism
CONTENT : a portrait of a man inside a frame

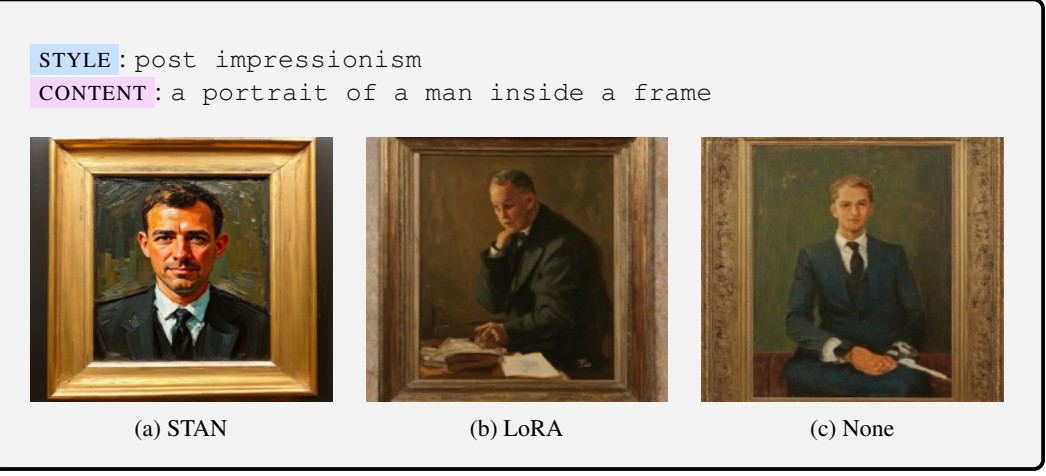

(a) STAN          (b) LoRA          (c) None

STYLE : pre raphaelite brotherhood
CONTENT : three female figures{dancing and playing

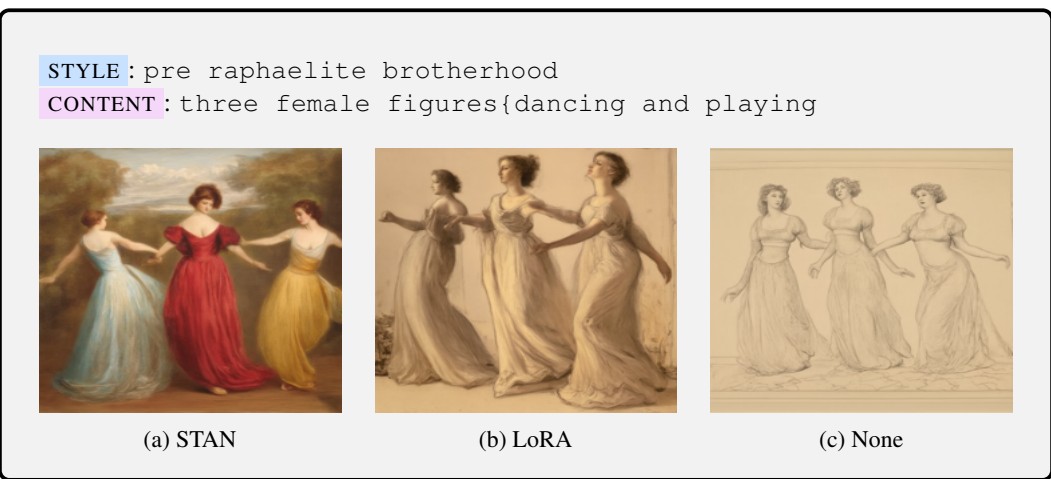

(a) STAN          (b) LoRA          (c) None

STYLE : realism
CONTENT : a woman

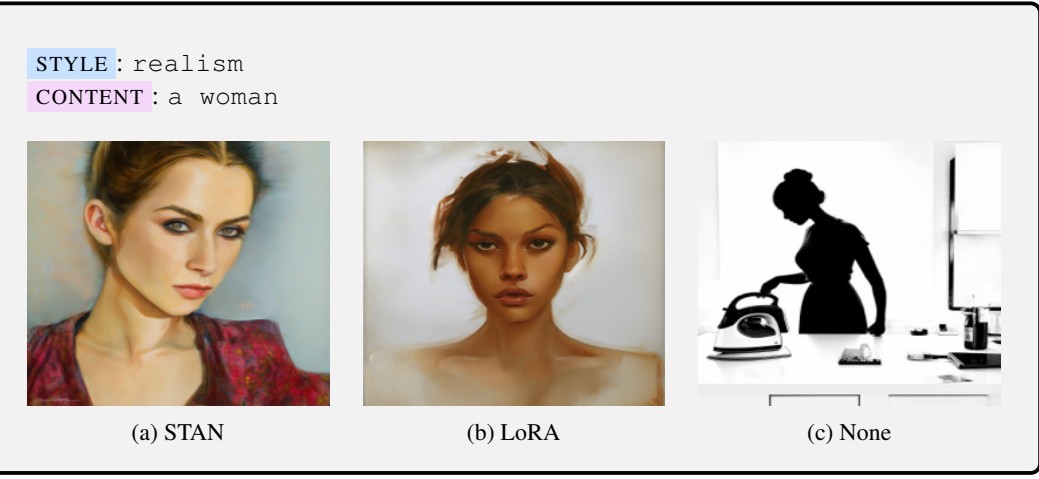

(a) STAN          (b) LoRA          (c) None

STYLE : rococo
CONTENT : a man

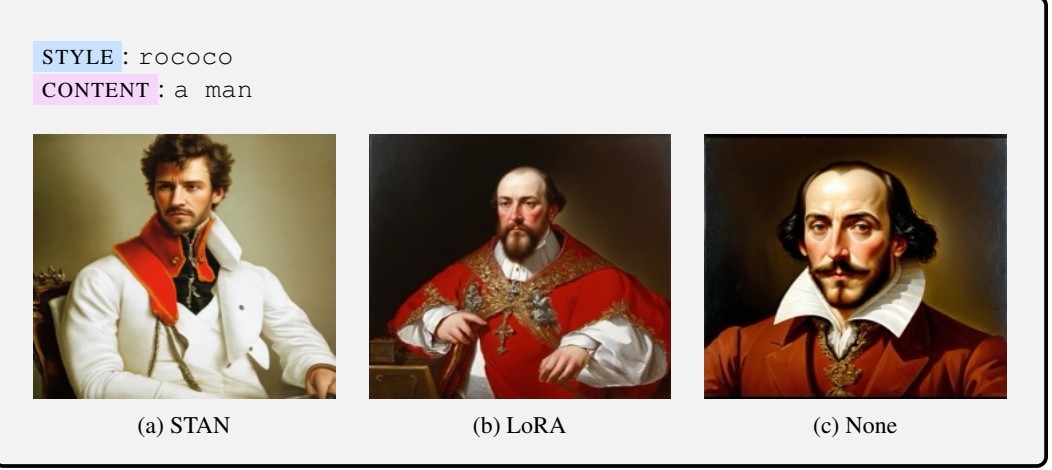

(a) STAN          (b) LoRA          (c) None

STYLE : romanticism
CONTENT : a woman and a white horse

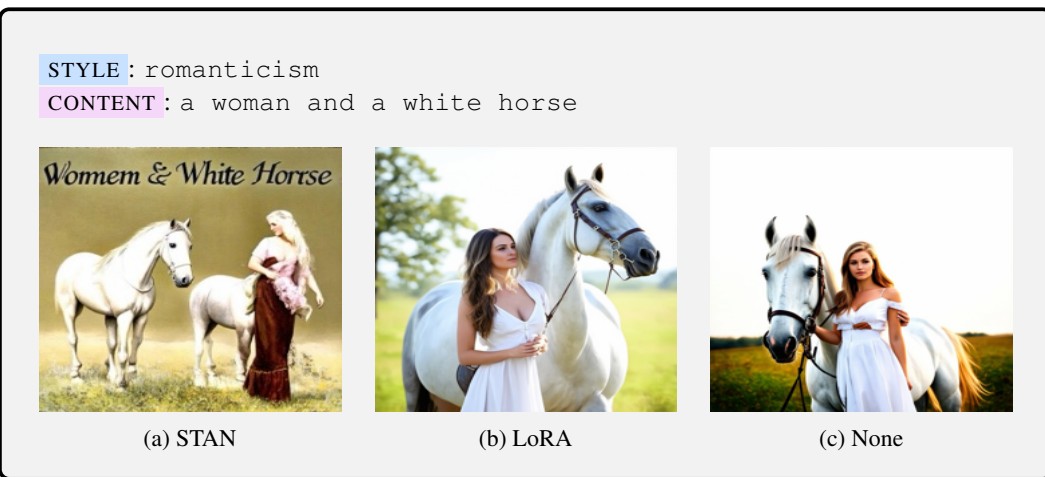

(a) STAN (b) LoRA (c) None

STYLE : surrealism
CONTENT : blancs

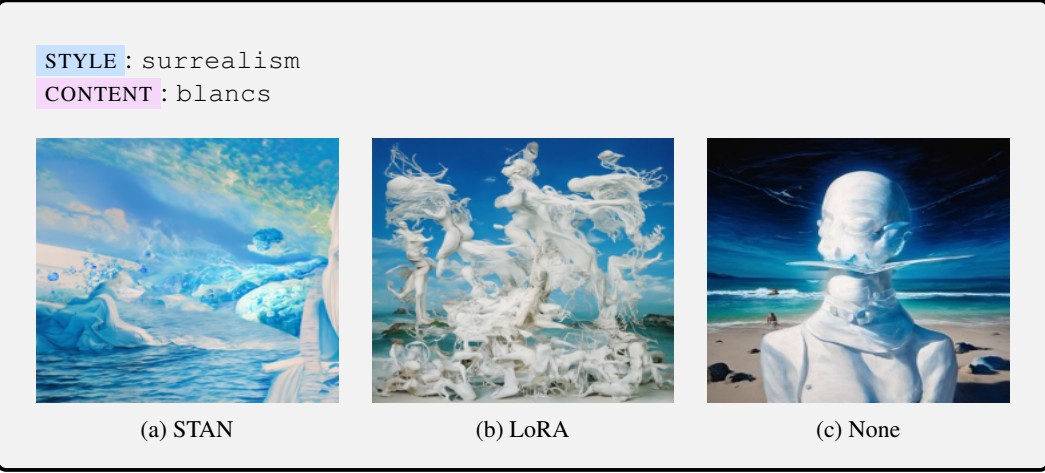

(a) STAN (b) LoRA (c) None

STYLE : symbolism
CONTENT : st-anthony

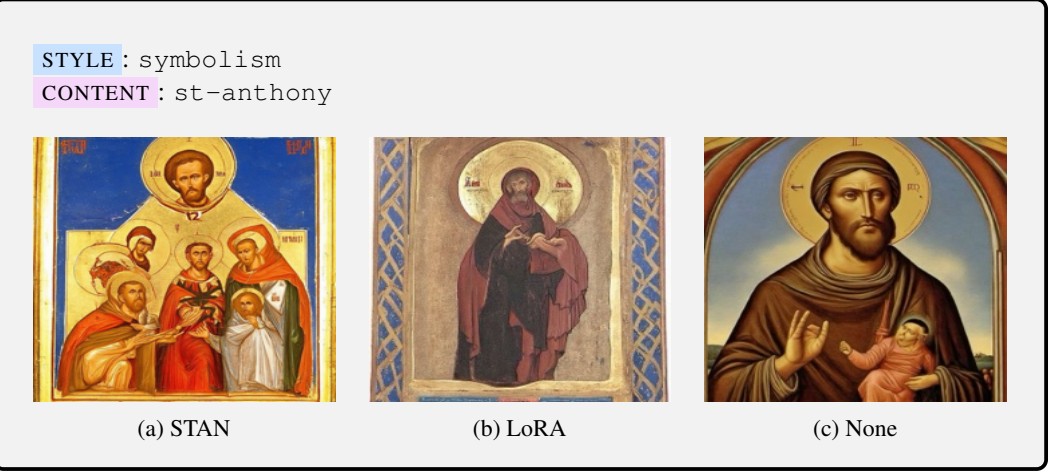

(a) STAN (b) LoRA (c) None

STYLE : ukiyoe
CONTENT : Mount Fuji behind blooming cherry trees

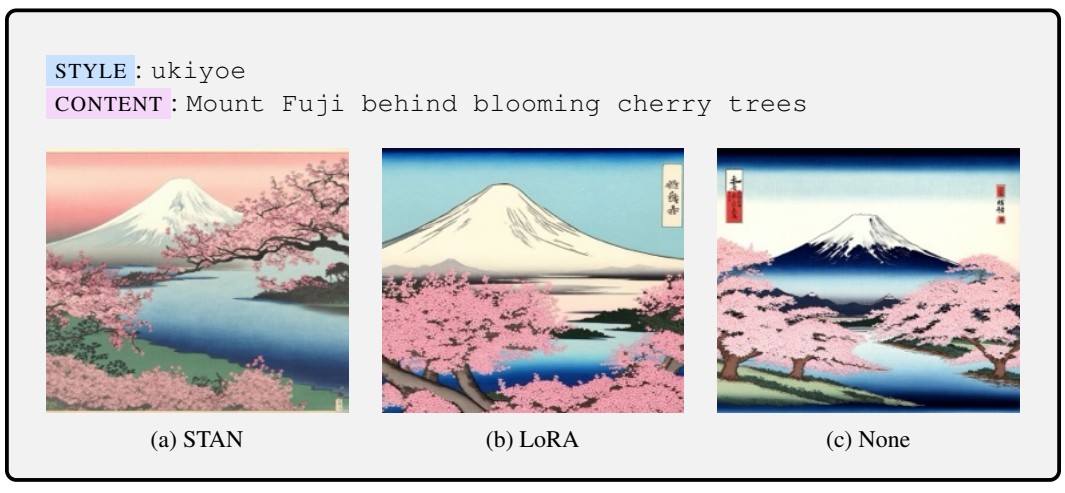

(a) STAN          (b) LoRA          (c) None

# F STATISTICAL ANALYSIS OF SPARSE LATENT REPRESENTATIONS

In this section, we conduct a statistical analysis of the sparse latent representations learned by STAN, as visualized in Figure 3. Our goal is to quantify the behavior and specialization of individual latent units by measuring their activation patterns across different input samples and semantic classes. Specifically, we evaluate three statistical properties for each latent unit: the frequency of activation (x-axis), the mean activation value (y-axis), and the label entropy (color bar), which reflects the class specificity of each latent.

Let $\{a_i\}_{i=1}^N$ denote the activation values of a given latent unit across $N$ input samples. We consider a latent unit to be *active* on sample $i$ if $a_i > 0$. Let $\mathcal{I} = \{i \mid a_i > 0\}$ represent the index set of active samples, and let $N+ = |\mathcal{I}|$ be the number of activations. Each input sample is associated with a ground truth class label $y_i \in \mathcal{C}$, where $\mathcal{C}$ denotes the set of all class labels.

**(1) Mean Activation Frequency.** We compute the mean frequency with which the latent unit is activated across the dataset as:

$$\text{MeanFreq} = \frac{1}{N} \sum_{i=1}^N \mathbf{1}[a_i > 0] = \frac{N_+}{N}. \tag{18}$$

This metric reflects how often the latent contributes non-zero activation to the representation.

**(2) Mean Activation Value.** For the subset of samples where the latent is active, we compute the average activation magnitude:

$$\text{MeanAct} = \frac{1}{N_+} \sum_{i \in \mathcal{I}} a_i. \tag{19}$$

This value captures the typical strength of activation, conditioned on the latent being active.

**(3) Label Entropy.** To assess how class-specific the latent activation is, we compute a label-weighted activation distribution. First, we define the activation proportion $p_c$ for each class $c \in \mathcal{C}$ as:

$$p_c = \frac{\sum_{i:, y_i = c} a_i, \mathbf{1}[a_i > 0]}{\sum_{j=1}^N a_j, \mathbf{1}[a_j > 0]}. \tag{20}$$

Then, we compute the entropy of this distribution:

$$\text{Entropy} = -\sum_{c \in \mathcal{C}} p_c \log p_c. \tag{21}$$

Lower entropy values indicate that the latent is primarily activated by a narrow set of classes, implying class specialization. Conversely, higher entropy suggests that the latent is shared across multiple classes, capturing more general or abstract features.

These three metrics together allow us to characterize both the usage pattern and semantic specificity of each latent dimension in STAN. Additional examples and visualizations of these statistics are provided in Appendix G.

# G   MORE VISUALIZATIONS FOR DIFFERENT STAN INJECTIONS

In this section, we present additional visualizations consistent with those shown in Figure 3, constructed using the statistical analysis formulations described in Appendix F. These supplementary results provide further evidence that both sparsity and broad representational capacity are consistently exhibited across different injection layers within the model. This consistency indicates that the behavior of STAN is not limited to a specific configuration, but instead reflects a generalizable property of the method. The observed uniformity and stability across injections support the reliability and robustness of STAN.

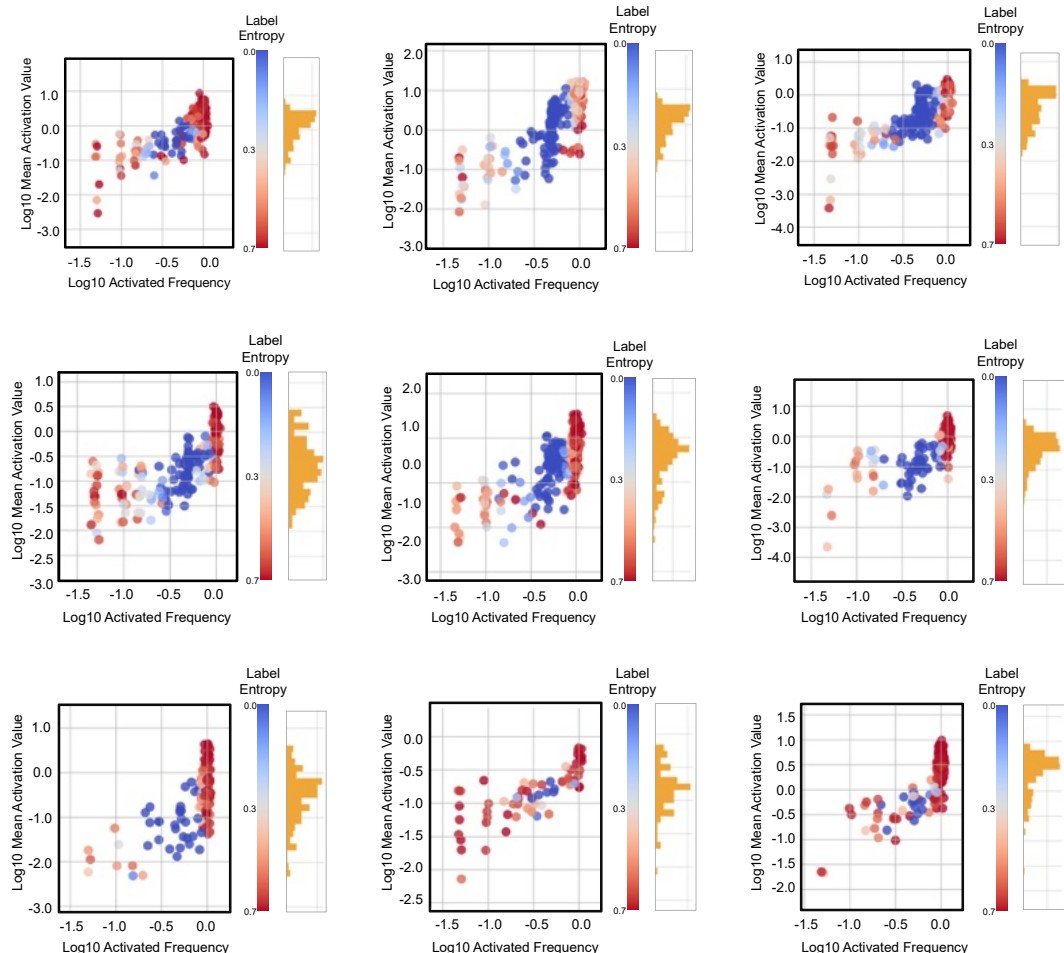

Figure 32: **Visualization examples of sparse representations across different STAN injection layers.** Each subplot presents the distribution of middle-layer feature activations, with both axes scaled using $\log_{10}$ to improve visual clarity. The color bar denotes label entropy, where lower values indicate class-specific activations and higher values reflect class-general behavior, as defined in Appendix F. To the right of each plot, a yellow bar chart illustrates the average activation magnitude for each unit, providing additional insight into the strength and distribution of sparsely activated features.

## H  INTERPRETING DISENTANGLED FEATURES IN LLM

In this section, we present additional interpretability experiment on language model utilizing the similar setting shown in Figure 3 and Appendix F.

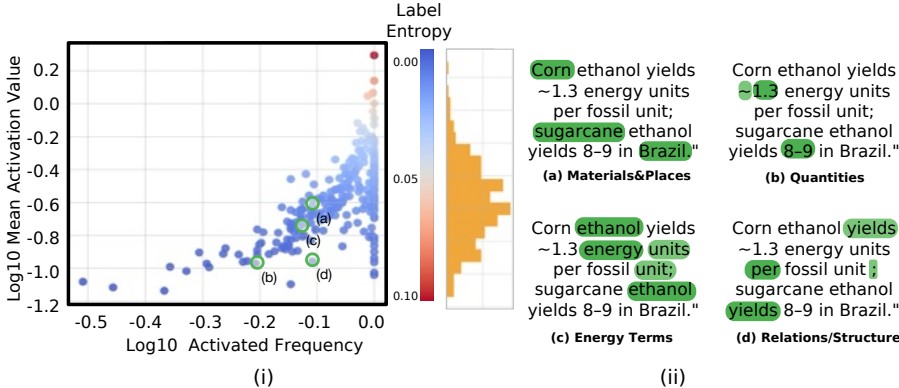

Figure 33: Qualitative demonstration of the sparse representations learned by STAN. (i) Distribution of sparsely activated intermediate features across four distinct styles. (ii) Highlight of the corresponding text concepts: (a) Materials & Places, (b) Quantities, (c) Energy Terms and (d) Pelations / Structure.

Part (i) illustrates activation frequency statistics of sparsely activated intermediate features when model is fed with different texts. The y-axis represents $\log_{10}$ mean activation value, and the x-axis indicates $\log_{10}$ activation frequency. Each point is colored according to its label entropy, which reflects the degree of concept specificity associated with that neuron: lower entropy values indicate specialization (i.e., the neuron is primarily activated by one concept), while higher entropy suggests activation by a broader mix of styles. Details on the formulation of this analysis can be found in Appendix F. Part (ii) presents corresponding visual examples, illustrating that the texts activating a given neuron are consistently aligned with its associated concept. The results reveal that different concepts tend to activate distinct, often non-overlapping, subsets of sparse latent features. A small number of shared neurons appear to capture common generative priors, while the majority remain concept-specific. It demonstrates the expressive capacity of subspace combinations in the sparse latent space.

## I  THE USE OF LARGE LANGUAGE MODELS

In line with the ICLR policy, we disclose the use of Large Language Models during the preparation of this manuscript. Our use of these tools was strictly limited to assistance with language and formatting. Specifically, we employed an LLM for proofreading, correcting grammatical errors, and improving the clarity and readability of sentences. The LLM had no role in the core scientific aspects of this work, including research ideation, methodological design, experimental analysis, or the generation of any results or conclusions. All intellectual contributions and the core content of this paper are solely the work of the authors.

