# OpenReview forum: "The Chosen Few: Sparse Adaptation for Large Models"
_ICLR.cc/2026/Conference — Submitted to ICLR 2026_

### Official Review · Reviewer_8rX2 · 2025-10-15

**Soundness:** 2
**Presentation:** 1
**Contribution:** 2
**Rating:** 2
**Confidence:** 4

**Summary:**

The manuscript under review suggests an improvement over LoRA (which packs task changes into a tiny, dense, low-rank space) by proposing STAN, which drops tiny sparse autoencoder (SAE) modules into a frozen model and lets only a few latent “neurons” fire per input (a Top-K selection). Concretely, the encoder maps activations into a large latent space, the Top-K gate keeps only the strongest features, and the decoder maps them back to produce the adjustment added to the frozen layer’s output. The methodology is easy to follow (see sec 3). Across language (GLUE with RoBERTa/DeBERTa), multimodal VLMs (LLaVA), and a diffusion model (Stable Diffusion 3), STAN matches PEFT baselines like LoRA/AdaLoRA/SoRA. However, in the crowded world of LLM finetuning after 2022 (LoRA), the experimental section is lacking and also I feel there are some caveats related to the results (see weaknesses)

**Strengths:**

1. Clear, intuitive idea: Swapping “low-rank and dense” for “high-dimensional and sparse” is well-motivated, and the Top-K SAE view makes the adaptation non-linear and input-specific (a mix of subspaces instead of one fixed subspace). The math is compact and readable.
2. Broad, consistent wins. On GLUE and several LLaVA benchmarks, STAN seems to perform well.
3. Generative demo with interpretability. The SD3 “style alignment” experiment shows both better scores (CLIP/DINO) and a convincing story that different styles trigger different sparse units—nice evidence of disentanglement.
4. Positioning vs. LoRA-family. The critique of LoRA’s rigidity/entanglement and the argument why this method helps are clearly laid out.

**Weaknesses:**

First of all to justify the validity of the technique, more experiments are needed especially applying it to different LLMs and instruction following (Mistral, Llama and GPT models are standard). Next, I don't understand from where the authors took the values of Table 1 and Table 2? did they reproduce the experiments? If they reproduced the experiments then why is that not mentioned in the tables as is standard? The only indication seems to be their declaration of the hyperparameters, but that is indirect and also looking at the appendices is a choice not an obligation for the reviewers. At first I thought the values are standard but then I had some doubts and I rechecked and some of the reported values seem to be lower than in the VeRA paper (ICLR 2024). There may exist other works in the meantime which reported higher scores. For Table 4 LoRA, where are the scores reported from?
Further,

1. The paper says the SAE path doesn’t increase inference complexity, but practical costs (latency/throughput/memory on real deployments) aren’t quantified; E/D projections and Top-K still add ops, so clearer wall-clock and memory numbers would help.
2. Performance depends on K and latent expansion; ablations are on an MNLI subset only. It’s unclear how robust those settings are across tasks and scales without more sweeping studies.
3. The Top-K router is the core mechanism; training stability is said to be good, but implementation/tricks (e.g., straight-through choices, tie-breaking) are not discussed in detail in the main body.
4. CLIP/DINO scores are narrow; human evals, more styles, safety/control stress-tests would solidify the generative claims.

**Questions:**

See the weaknesses above.

---

> ### Author Response · Authors · 2025-11-20
>
> We sincerely thank the reviewer for the detailed and critical feedback, as well as for the positive assessment of our main ideas and results. We especially appreciate the recognition that (1) the “high-dimensional and sparse” perspective is intuitive and well-motivated, (2) STAN delivers broad and consistent improvements across GLUE and LLaVA benchmarks, (3) the SD3 style-alignment experiment provides a convincing disentanglement story, and (4) our positioning with respect to LoRA-family methods is clearly articulated. Below we address each concern point by point.
>
> ## W1: Scope of experiments, base models, and source of numbers in Tables
>
> (a) Breadth of models and tasks already evaluated
>
> We agree that validating a new PEFT technique on diverse architectures is important. We would like to clarify that STAN has already been evaluated on multiple model families, modalities, and task types, not just a single LLM:
>
> 1. Tables 1–2 evaluate STAN on RoBERTa-base/large and DeBERTaV3-base/large across five GLUE tasks (MNLI, SST-2, QNLI, QQP, CoLA), showing consistent gains over LoRA, AdaLoRA, SoRA, and a large set of additional PEFT baselines.
>
> 2. Table 4 evaluates STAN on LLaMA-2-7B for GSM8K and MATH, again outperforming LoRA and PiSSA.
>
> 3. Tables 3 and 5 evaluate STAN on three LLaVA variants (LLaVA-1.5-7B, LLaVA-1.5-13B, LLaVA-1.6-Vicuna-13B) on GQA, ScienceQA, VizWiz, POPE, and MMBench, all instruction-following, vision-language benchmarks.
>
> 4. Section 4.2 and the appendix further extend STAN to Stable Diffusion 3, showing both quantitative (CLIP/DINO) and qualitative style alignment improvements.
>
> Taken together, these experiments cover encoder LMs, decoder LMs, instruction-tuned VLMs, and diffusion generators, suggesting that STAN is architecture-agnostic and robust across modalities.
>
> We agree that including additional base models and newer instruction models would further strengthen the story. Due to the paper scope limitation, we focused on a representative but still diverse subset.
>
>
> (b) Where do the values in Tables come from?
>
> 1. We apologize for the lack of explicit clarification in the current draft. All baseline numbers in Tables 1, 2, 3, 4, and 5 are from our own re-runs, not copied from prior papers. For each method we use the official code following the hyperparameters recommended in the original papers and run them under a unified training environment and seed, as summarized in our implementation details.
>
> 2. The reason some reported baselines are slightly lower than those in their original paper is because we re-ran everyone in the same environment instead of mixing numbers reported under different training setups. This avoids unfair advantages/disadvantages due to varying hardware, seeds, or training schedules. We did not directly copy numbers from other papers, since it's irresponsible.
>
> 4. We agree that the current tables do not make this explicit. In the revised version, we will add a clear note in the Experimental Setup section:
>      > “All results are reproduced by us in a unified training environment using official implementations.”
>
> We thank the reviewer for pressing us to clarify this; we believe this change will make the comparisons more transparent and trustworthy.

---

> ### Author Response · Authors · 2025-11-20
>
> ## W2: The paper says the SAE path doesn’t increase inference complexity, but practical costs (latency/throughput/memory on real deployments) aren’t quantified; E/D projections and Top-K still add ops, so clearer wall-clock and memory numbers would help.
>
> 1. STAN indeed introduces additional computation due to the Top-K selection. We will update the text to state that STAN " with only modest overhead addition" rather than “no additional complexity”.
>
> 2. We report inference latency, throughput, GPU-hours, and peak memory in our runtime analysis in Appendix C.1. On RoBERTa-base, compared with AdaLoRA and SoRA: STAN has comparable inference latency but much higher training throughput and fewer GPU-hours. On LLaVA-1.5-13B, compared with LoRA: STAN has similar per-sample inference latency, higher training throughput, about half the GPU-hours for the same 3-epoch run and only a modest increase in peak memory.
>
> 3. Further, we additionally statistic the effective tunable parameters in comparison with LoRA, here is the results.
>
>    | Model              | Adapter     | Effective Tunable Parameters |
>    |--------------------|-------------|--------------------------------|
>    | RoBERTa-base       | STAN        | 1.3M                  |
>    | RoBERTa-base       | LoRA        | 1.9M                      |
>    | RoBERTa-large      | STAN        | 3.5M                  |
>    | RoBERTa-large      | LoRA        | 4.5M                      |
>    | DeBERTaV3-base     | STAN        | 1.3M                  |
>    | DeBERTaV3-base     | LoRA        | 1.9M                      |
>    | DeBERTaV3-large    | STAN        | 3.5M                  |
>    | DeBERTaV3-large    | LoRA        | 4.5M                     |
>
>    Although the SAE contains a high-dimensional encoder/decoder, only a small subset of latent unit are active at each forward pass due to sparsity. Across all encoder models, STAN consistently uses fewer effective finetuning parameters than LoRA, despite achieving stronger performance. Since sparsity ensures that STAN only updates a small, input-dependent subset of features, the actual computational footprint and parameter usage during finetuning are actually reduced compared to dense low-rank adapters. This further supports our claim that STAN achieves excellent performance while maintaining strong parameter efficiency.
>
> These results show that while we do pay a small runtime cost on the STAN, the practical overhead is modest, especially in large models where backbone computation dominates. At the same time, training becomes more efficient due to sparsity.
>
> ## W3: Performance depends on K and latent expansion; ablations are on an MNLI subset only. It’s unclear how robust those settings are across tasks and scales without more sweeping studies.
>
> Following this suggestion, we conducted additional ablations on more GLUE task subsets to test robustness of the Top-K and expansion factor across tasks.
>
> ### a) SST-2 (accuracy)
>
> | Top-K | Exp=1  | Exp=1.5 | Exp=2  |
> |-------|--------|---------|--------|
> | 1     | 0.9369 | 0.9358  | 0.9346 |
> | 4     | 0.9323 | 0.9346  | 0.9427 |
> | 8     | 0.9346 | 0.9404  | 0.9438 |
> | 16    | 0.9381 | 0.9335  | 0.9450 |
> | 32    | 0.9415 | 0.9392  | 0.9392 |
> | 64    | 0.9404 | 0.9369  | 0.9392 |
>
> ### (b) CoLA (Matthews corr.)
>
> | Top-K | Exp=1  | Exp=1.5 | Exp=2  |
> |-------|--------|---------|--------|
> | 1     | 0.5495 | 0.5855  | 0.5730 |
> | 4     | 0.5599 | 0.5701  | 0.5727 |
> | 8     | 0.5651 | 0.5727  | 0.5677 |
> | 16    | 0.5779 | 0.5598  | 0.5786 |
> | 32    | 0.5730 | 0.5778  | 0.5829 |
> | 64    | 0.5729 | 0.5702  | 0.5856 |
>
> ### c) STSB (accuracy)
>
> | Top-K | Exp=1  | Exp=1.5 | Exp=2  |
> |-------|--------|---------|--------|
> | 1     | 0.8655 | 0.8616  | 0.8716 |
> | 4     | 0.8618 | 0.8648  | 0.8614 |
> | 8     | 0.8603 | 0.8609  | 0.8610 |
> | 16    | 0.8777 | 0.8956  | 0.8763 |
> | 32    | 0.8776 | 0.8704  | 0.8667 |
> | 64    | 0.8963 | 0.8670  | 0.8768 |
>
> ### d) MRPC (accuracy)
>
> | Top-K | Exp=1  | Exp=1.5 | Exp=2  |
> |-------|--------|---------|--------|
> | 1     | 0.8000 | 0.8000  | 0.8020 |
> | 4     | 0.8232 | 0.8130  | 0.8123 |
> | 8     | 0.8218 | 0.8191  | 0.8157 |
> | 16    | 0.8157 | 0.8061  | 0.8061 |
> | 32    | 0.8089 | 0.8041  | 0.8034 |
> | 64    | 0.8061 | 0.8020  | 0.8027 |
>
> ### e) RTE (accuracy)
>
> | Top-K | Exp=1  | Exp=1.5 | Exp=2  |
> |-------|--------|---------|--------|
> | 1     | 0.8524 | 0.8376  | 0.7860 |
> | 4     | 0.8303 | 0.8118  | 0.8192 |
> | 8     | 0.8044 | 0.8044  | 0.8118 |
> | 16    | 0.8081 | 0.8155  | 0.8118 |
> | 32    | 0.8118 | 0.8044  | 0.7970 |
> | 64    | 0.8044 | 0.8118  | 0.8081 |
>
> Across all tasks, performance is quite stable over a wide range of (Top-K, expansion) settings; differences are generally small rather than catastrophic. Importantly, even at high sparsity, STAN retains or improves performance, demonstrating that the method is robust to hyperparameter choices and benefits from sparsity rather than being overly sensitive.

---

> ### Author Response · Authors · 2025-11-20
>
> ## W4: The Top-K router is the core mechanism; training stability is said to be good, but implementation/tricks (e.g., straight-through choices, tie-breaking) are not discussed in detail in the main body.
>
> 1. We follow an implementation where the encoder produces a high-dimensional latent vector, we apply Top-K on activations, and then decode. Gradients are passed using a straight-through for the Top-K mask, without any additional tricks such as custom tie-breaking schemes beyond the Top-K behavior.
>
> 2. As for the training stability, we provide loss curves across models and tasks in Appendix D, comparing STAN to LoRA, AdaLoRA, and SoRA. STAN consistently exhibits smooth, monotonic convergence with low variance. SoRA, although also sparse, shows significant oscillations and sometimes divergence. LoRA/AdaLoRA converge more stably than SoRA but typically to higher final losses than STAN. These results support our claim that STAN is stable in practice.
>
> 3. The stability arises from (1) the simplicity of the training objective (no extra self-supervision; STAN is trained directly with the downstream task loss, as in standard PEFT), (2) the fact that Top-K selects a small but consistent subset of features per input, and (3) the absence of aggressive pruning or proximal updates that can destabilize training.
>
> ## W5: CLIP/DINO scores are narrow; human evals, more styles, safety/control stress-tests would solidify the generative claims.
>
> We thank the reviewer for this excellent suggestion. We agree that purely automatic metrics like CLIP/DINO, while useful, provide a narrow view of generative quality and alignment. Following the reviewer’s recommendation, we conducted an additional user study focusing on style alignment and image quality in the SD3 experiments:
>
> 1.  We consider three methods: None (pretrained SD3 without finetuning), LoRA, and STAN. For each participant, we randomly sample 20 style prompts, and for each prompt we show three images A/B/C (corresponding to the three methods, order randomized). For each triple, users answer the question: “Which image best matches the target style and has the highest overall visual quality?” We aggregate win rate per method across all comparisons in the following table.
>
>        | Method | Wins | Total | Win Rate |
>        |--------|------|-------|----------|
>        | None   | 6    | 501   | 1.20%    |
>        | LoRA   | 39   | 501   | 7.78%    |
>        | **STAN (Ours)** | **456** | **501** | **91.02%** |
>
>        STAN is preferred in over 90% of the comparisons, both in terms of style alignment and perceived image quality, significantly outperforming both the pretrained model and the LoRA-finetuned model.
>
> 2. These results strongly corroborate the CLIP/DINO scores in our main paper, showing that the improvements are not only numerical but also perceptually meaningful to human users. Combined with the disentanglement analysis in our style experiments, they strengthen the claim that STAN learns disentangled, style-specific sparse features that support more precise and controllable generation. We have add this additional user study to our revised paper in Section 4.2.
>
> 3. We agree that exploring safety and control stress-tests (e.g., robustness to adversarial prompts or harmful content) is an important direction. In this work, we focused primarily on style controllability and interpretability rather than safety.
>
> ---
>
> In short, we sincerely thank the reviewer for providing insightful and constructive comments. We are also open to further discussion for improving the quality of our paper.

---

> > ### Author Response · Authors · 2025-11-28
> >
> > Dear Reviewer 8rX2,
> >
> > Hopefully we have fully addressed your concern in the response listed top. If you find it satisfactory, we respectivefully ask you to re-evaluate the rating. And please do not hesitate to let us know if there is any further questions!
> >
> > Thanks and wish you all the best!
> >
> > Authors

---

### Official Review · Reviewer_gQSZ · 2025-10-30

**Soundness:** 2
**Presentation:** 3
**Contribution:** 3
**Rating:** 4
**Confidence:** 3

**Summary:**

This paper proposes replacing additive PEFT methods, such as LoRA, with SAE to improve model performance. The authors validate the superiority of the proposed approach over baselines on language models, VLMs, and SD models.

**Strengths:**

- The overall writing of the paper is clear and easy to read.
- The authors provide a comprehensive set of tasks to demonstrate the superiority of their method.
- The experiments in Fig. 3 can, to some extent, illustrate the role of high-dimensional sparsity in concept disentanglement during fine-tuning.

**Weaknesses:**

- Many sparse fine-tuning methods are not discussed or compared.[1-4]
- Line 98 claims that no additional inference complexity is introduced, but incorporating SAE does require extra computational overhead.
- The experimental section lacks many details.
    - The study on interpretability brought by sparsity is limited to Fig. 3, and this experiment does not provide the performance of comparative methods.
    - What is the training procedure of the model? Does introducing SAE require an additional self-supervised training process? What is the associated training cost?
    - How is SAE specifically integrated into the model? How does its placement affect the model’s performance?
    - The comparative experiments do not report the tunable parameter scales for the different methods.

[1] Sensitivity-aware visual parameter-efficient fine-tuning. CVPR'23

[2] Gradient-based parameter selection for efficient fine-tuning. CVPR'24

[3] On the effectiveness of parameter-efficient fine-tuning. AAAI'23

[4] Expanding Sparse Tuning for Low Memory Usage. NeurIPS'24

**Questions:**

- How is SAE specifically integrated into the model? Is it used only once or multiple times? The authors should provide detailed explanations along with comparative experiments.

---

> ### Author Response · Authors · 2025-11-20
>
> We sincerely thank the reviewer for the thoughtful evaluation and for highlighting the strengths of our work. We appreciate the recognition that (1) the paper is clearly written, (2) the empirical evaluation covers a broad range of tasks, and (3) the experiments (e.g., Fig. 3) provide evidence that high-dimensional sparsity helps with concept disentanglement during finetuning. Below we address the reviewer’s concerns point by point.
>
> ## W1: Many sparse fine-tuning methods are not discussed or compared. [1–4]
>
> We appreciate the reviewer for bringing these related works to us. We will revise the related work section to discuss these methods explicitly and clarify their relationship to STAN.
>
> 1. The cited works [1–4] share a high-level connection with STAN in that they all explore parameter-efficient or sparse adaptation. However, they largely operate in a different regime: they focus on selecting or sparsifying subsets of existing parameters (e.g., sensitivity-aware selection, gradient-based selection, or sparse tuning of weights) rather than learning sparse latent features. In contrast, STAN performs adaptation in a high-dimensional latent space and uses input-dependent Top-K sparsity to dynamically activate different feature subsets, which is a different form of sparsity.
>
> 2. Methodological differences.
>    - Sensitivity-aware visual PEFT [1] and gradient-based parameter selection [2] focus on which parameters of the backbone are most effective to tune, often in vision settings.
>    - On the effectiveness of PEFT [3] mainly provides a systematic empirical study of various PEFT strategies, most of which are low-rank or dense in the latent space.
>    - Expanding Sparse Tuning [4] emphasizes sparse tuning for low memory usage, still in the space of existing weights.
>    - STAN, instead, focuses on sparse, interpretable latent features learned by an SAE, which enables dynamic, input-dependent adaptation and concept-level analysis.
>
>    We view these methods as largely complementary: they sparsify or select in parameter space, whereas STAN introduces sparsity in a learned latent concept space. And we have added these works in our revised related work.
>
> ## W2: Line 98 claims that no additional inference complexity is introduced, but incorporating SAE does require extra computational overhead.
>
> We thank the reviewer for pointing out this. We agree that the statement at Line 98 is a slight over-claim and we will revise the wording accordingly.
>
> 1. STAN indeed introduces additional computation due to the Top-K selection. We have updated the text to state that STAN " with only modest overhead addition" rather than “no additional complexity”.
>
> 2. The overhead is small and diminishes with model size. As shown in Appendix C.1, the input-dependent pathway adds minimal latency in practice, and importantly, the relative overhead decreases as model size increases. On larger models (e.g., 13B-scale multimodal LLMs), STAN’s inference latency approaches that of LoRA.
>
> 3. Training efficiency advantage. Appendix C.1 also reports training token throughput, GPU hours, and peak GPU memory. Due to the computational efficiency of sparsity in the latent activations, STAN can achieve overall better training throughput than other dense or adaptive-rank baselines.
>
> In summary, we acknowledge the reviewer’s concern and revised the claim in the main text. At the same time, our empirical results show that the added complexity is modest, increasingly negligible for large models, and improved upon training efficiency.
>
> ## W3: The study on interpretability brought by sparsity is limited to Fig. 3, and this experiment does not provide the performance of comparative methods.
>
> 1. We thank the reviewer for this suggestion and agree that interpretability is an important aspect of our method. Beyond Fig. 3 (style alignment), we also present language-model interpretability experiments in Appendix H, where we analyze how different sparse latent units correspond to distinct semantic concepts. These results support the claim that STAN’s sparse features are disentangled and concept-specific.
>
> 2. We agree that including comparative methods (e.g., LoRA) in the interpretability study would strengthen the story. During the rebuttal phase, we run additional experiments where we apply LoRA under the same setup and probe its latent directions using the same attribution and activation-frequency analysis. We have added this to revised paper in Section 4.2 Our findings suggest that dense low-rank adaptations lead to more entangled features compared to STAN’s sparse latent units.

---

> ### Author Response · Authors · 2025-11-20
>
> ## W4: What is the training procedure of the model? Does introducing SAE require an additional self-supervised training process? What is the associated training cost?
>
> 1. STAN follows a standard supervised PEFT pipeline. We start from a frozen pretrained backbone, then attach the STAN modules to target layers. We finetune only the STAN parameters using the task loss of the downstream dataset. No additional self-supervised pretraining or auxiliary loss is required for STAN. The encoder and decoder are trained jointly with the downstream objective, which keeps the method straightforward and easy to integrate. In the revised main paper, we will add a concise explanation that STAN does not require an extra self-supervised SAE pretraining phase.
>
> 2. For training cost and overhead, as detailed in Appendix C.1, we report: (1) GPU hours for finetuning; (2) Training token throughput and (3) Peak GPU memory usage. Due to the sparse activations, STAN enjoys computational savings, making its training cost competitive with or better than baselines.
>
> ## W5: How is SAE specifically integrated into the model? How does its placement affect the model’s performance?
>
> 1. Integration pattern. For Transformer-based backbones, we follow standard LoRA-style placement and attach STAN modules to the attention projections. Specifically, we place STAN on the target linear projection matrices in each Transformer block. Each such module takes the local hidden states as input, passes them through the encoder, applies Top-K sparsification, and then decodes the sparse latent into an additive adaptation.
>
> 2. The STAN is therefore used multiple times, once per STAN-augmented projection in each Transformer layer where we enable adaptation. We do not use a single global SAE shared across the entire network; instead, the adapters are localized to specific layers and projections.
>
> 3. Effect of placement. We followed common PEFT practice (e.g., LoRA) and found that adapting linear projections offers a good trade-off between expressivity and parameter/compute efficiency. Placing STAN on these target projections already yields strong performance, so we adopte this as the default configuration for simplicity and fairness, make sure that all the comparison are using the same configuration.
>
> ## W6: The comparative experiments do not report the tunable parameter scales for the different methods.
>
> 1. We thank the reviewer for pointing out this missing detail. Below we provide a clear accounting of the effective tunable parameters for both STAN and LoRA.
>
>    | Model              | Adapter     | Effective Tunable Parameters |
>    |--------------------|-------------|--------------------------------|
>    | RoBERTa-base       | STAN        | 1.3M                  |
>    | RoBERTa-base       | LoRA        | 1.9M                      |
>    | RoBERTa-large      | STAN        | 3.5M                  |
>    | RoBERTa-large      | LoRA        | 4.5M                      |
>    | DeBERTaV3-base     | STAN        | 1.3M                  |
>    | DeBERTaV3-base     | LoRA        | 1.9M                      |
>    | DeBERTaV3-large    | STAN        | 3.5M                  |
>    | DeBERTaV3-large    | LoRA        | 4.5M                     |
>
>    Although the SAE contains a high-dimensional encoder/decoder, only a small subset of latent unit are active at each forward pass due to sparsity. Across all encoder models, STAN consistently uses fewer effective finetuning parameters than LoRA, despite achieving stronger performance.
>
> 2. Since sparsity ensures that STAN only updates a small, input-dependent subset of features, the actual computational footprint and parameter usage during finetuning are actually reduced compared to dense low-rank adapters. This further supports our claim that STAN achieves excellent performance while maintaining strong parameter efficiency.
>
>
>
> ## Q1: How is SAE specifically integrated into the model? Is it used only once or multiple times?
>
> We thank the reviewer again for this question. This is closely related to W5 and W6, and we will clarify in more detail.
>
> 1. STAN is integrated into the backbone by attaching adapters to target linear projections of each Transformer block and is used multiple times throughout the network, once per adapted projection per block, rather than as a single global module.
>
> 2. The training procedure remains fully aligned with standard PEFT. The backbone is frozen and only the STAN modules are trained under the downstream task loss without any additional self-supervised pretraining. Training cost and resource usage are reported in Appendix C.1, and are competitive due to sparsity. We also reported the effective tunable parameter comparison with LoRA in the respond to W6.
>
> ---
>
> In short, we sincerely thank the reviewer for providing insightful and constructive comments. We are also open to further discussion for improving the quality of our paper.

---

> > ### Comment · Reviewer_gQSZ · 2025-11-25
> >
> > I sincerely appreciate the authors for their response. However, I still have the remaining concerns.
> >
> > W2: The current cost analysis is neither rigorous nor sufficiently comprehensive. For example:
> > - How many tunable parameters are employed by the baselines/STAN under the reported settings (Appendix C.1)?
> > - How does STAN perform under different configurations of tunable parameter budgets?
> > - How is the Top-K operation efficiently implemented in your method? Given that Top-K selection over high-dimensional activations is often a computational bottleneck, it would be important to clarify how this issue is addressed in STAN.
> > - As the hidden dimension increases in larger models, the forward cost of the SAE inevitably grows as well. The authors are encouraged to provide a more rigorous scaling law (e.g. comparing STAN to LoRA) to support their claim, "importantly, the relative overhead decreases as model size increases."
> >
> > W2\&W4: The authors repeatedly claim that “due to the sparse activations, STAN enjoys computational savings.” However, it remains unclear how sparsity concretely contributes to reducing the computational cost. Could the authors clearly specify the mechanisms through which sparsity gives rise to actual savings during both forward and backward computation?
> >
> > W5: The authors should provide a detailed configuration table, specifying the placement and count of the SAE modules within the model.
> >
> > W6: "Since sparsity ensures that STAN only updates a small, input-dependent subset of features, the actual computational footprint and parameter usage during finetuning are actually reduced compared to dense low-rank adapters."
> > - All parameters in the SAE are trainable. Sparsity affects only the forward activations rather than the backward weight updates. Moreover, the activations of the SAE vary depending on the individual input samples. Therefore, counting only the sparsified activations while ignoring the full set of tunable SAE parameters is inappropriate and potentially misleading.
> > - If the authors employ sparse backpropagation strategies, which are typically unstructured, this would likely increase the training complexity. Could the authors clarify whether such methods are used in STAN, and if so, how they mitigate the additional training overhead?

---

> > > ### Author Response · Authors · 2025-11-26
> > >
> > > We sincerely thank the reviewer for the rigorous follow-up questions. We appreciate the opportunity to clarify our claims. .
> > >
> > > ## Q1: How many tunable parameters are employed by the baselines/STAN under the reported settings (Appendix C.1)?
> > >
> > > 1. The reviewer raises valid questions regarding the tunable parameter count. We clarify that our claim of "computational savings" refers specifically to **Training Throughput** and **Wall-clock Training Time**, which are the practical bottlenecks for practitioners, rather than static parameter storage.
> > >
> > > 2. Table below details the comparison under the reported settings of RoBERTa-base in Appendix C.1. While STAN maintains a larger number of total trainable parameters (Storage Cost) due to the high-dimensional encoder/decoder, this does not negatively impact training speed. On the contrary, due to the efficiency of sparse operations, STAN achieves higher throughput.
> > >
> > > | Method | Total Trainable Params (Storage) | Effective Params (Active/Fwd) | Training Throughput | Inference Latency |
> > > | :--- | :--- | :--- | :--- | :--- |
> > > | **AdaLoRA** | 1.9M | 1.9M | 13,721 tokens/sec | 23.29 ms |
> > > | **LoRA** | 1.9M | 1.9M | 12,338 tokens/sec | 23.18 ms |
> > > | **STAN (Ours)**| 5.7M | 1.3M | 19,815 tokens/sec | 23.16 ms |
> > >
> > >
> > > ## Q2: How does STAN perform under different configurations of tunable parameter budgets?
> > >
> > > We sincerely think the reviewer for this additional comparison suggestion. We evaluated STAN on GLUE tasks with varying Expansion Factors and Sparsity (Top-K selections). They respectively regulate the total budgets of parameters and the degree of sparsity (effective parameters). As shown in Table below, STAN exhibits high stability, the change of parameter budget does not drastically alter performance, and excellent results are achieved even with lower budgets.
> > >
> > > | Top-K | SST-2 (Exp=1) | SST-2 (Exp=1.5) | SST-2 (Exp=2) | CoLA (Exp=1) | CoLA (Exp=1.5) | CoLA (Exp=2) | STSB (Exp=1) | STSB (Exp=1.5) | STSB (Exp=2) | MRPC (Exp=1) | MRPC (Exp=1.5) | MRPC (Exp=2) | RTE (Exp=1) | RTE (Exp=1.5) | RTE (Exp=2) |
> > > |-------|---------------|-----------------|---------------|--------------|----------------|--------------|--------------|----------------|--------------|--------------|----------------|--------------|-------------|---------------|-------------|
> > > | 1     | 0.9369        | 0.9358          | 0.9346        | 0.5495       | 0.5855         | 0.5730       | 0.8655       | 0.8616         | 0.8716       | 0.8000       | 0.8000         | 0.8020       | 0.8524      | 0.8376        | 0.7860      |
> > > | 4     | 0.9323        | 0.9346          | 0.9427        | 0.5599       | 0.5701         | 0.5727       | 0.8618       | 0.8648         | 0.8614       | 0.8232       | 0.8130         | 0.8123       | 0.8303      | 0.8118        | 0.8192      |
> > > | 8     | 0.9346        | 0.9404          | 0.9438        | 0.5651       | 0.5727         | 0.5677       | 0.8603       | 0.8609         | 0.8610       | 0.8218       | 0.8191         | 0.8157       | 0.8044      | 0.8044        | 0.8118      |
> > > | 16    | 0.9381        | 0.9335          | 0.9450        | 0.5779       | 0.5598         | 0.5786       | 0.8777       | 0.8956         | 0.8763       | 0.8157       | 0.8061         | 0.8061       | 0.8081      | 0.8155        | 0.8118      |
> > > | 32    | 0.9415        | 0.9392          | 0.9392        | 0.5730       | 0.5778         | 0.5829       | 0.8776       | 0.8704         | 0.8667       | 0.8089       | 0.8041         | 0.8034       | 0.8118      | 0.8044        | 0.7970      |
> > > | 64    | 0.9404        | 0.9369          | 0.9392        | 0.5729       | 0.5702         | 0.5856       | 0.8963       | 0.8670         | 0.8768       | 0.8061       | 0.8020         | 0.8027       | 0.8044      | 0.8118        | 0.8081      |

---

> > > ### Author Response · Authors · 2025-11-26
> > >
> > > ## Q3: How is the Top-K operation efficiently implemented in your method? Given that Top-K selection over high-dimensional activations is often a computational bottleneck, it would be important to clarify how this issue is addressed in STAN.
> > >
> > > We sincerely appreciate the reviewer for raising the concern regarding the computational bottleneck of the Top-K selection mechanism. Our implementation leverages hardware accelerators available on the NVIDIA A100 GPUs used in our experiments and is supported by relative research.
> > >
> > > 1.  Hardware-Aware Optimization (NVIDIA A100):
> > >     * Memory-Bound Nature of Top-K: Top-K selection is fundamentally a memory-bound operation (involving scanning and sorting) rather than compute-bound. The A100 GPU provides significant advantages here: (1) High Bandwidth (HBM2e): The 1.6~2.0 TB/s memory bandwidth accelerates the scanning of high-dimensional activations. (2) Expanded L2 Cache: The A100 features a 40MB L2 cache (compared to 6MB in V100). This allows intermediate Top-K sorting results to reside largely on-chip, drastically reducing costly HBM round-trips.
> > >     * Sparse Acceleration: While our primary implementation uses custom kernels, the Ampere architecture is inherently optimized for sparse-dense operations. The computational overhead of the selection phase is negligible compared to the massive dense matrix multiplications (GEMMs) in the backbone model.
> > >
> > > 2.  Recent research [1, 2, 3] consistently demonstrates that in sparse/MoE architectures, the gating (Top-K) overhead is minimal relative to the savings gained by skipping computation for inactive latents/experts. And as shown in Table above, although the total staorage increased, this hardware-software synergy results in STAN achieving higher training throughput comparing to LoRA and AdaLoRA. The bottleneck of Top-K is theoretically present but empirically invisible in the end-to-end training loop on modern hardware.
> > >
> > > [1] Wen, T., et al. "Beyond Matryoshka: Revisiting Sparse Coding for Adaptive Representation." .
> > >
> > > [2] Gale, T., et al. "MegaBlocks: Efficient Sparse Training with Mixture-of-Experts." .
> > >
> > > [3] Fedus, W., et al. "Switch Transformers: Scaling to Trillion Parameter Models with Simple and Efficient Sparsity."
> > >
> > >
> > >
> > > ## Q4: As the hidden dimension increases in larger models, the forward cost of the SAE inevitably grows as well. The authors are encouraged to provide a more rigorous scaling law (e.g. comparing STAN to LoRA) to support their claim, "importantly, the relative overhead decreases as model size increases."
> > >
> > > We sincerely thank the reviwer for the additional experiment suggestion. To support our claim that "relative overhead decreases as model size increases," we provide a rigorous comparison of STAN versus LoRA and AdaLoRA across different model scales, combining data from Appendix C.1 with new experiments.
> > >
> > > 1. We conducted additional experiments on LLaVA-v1.5-7B (OCR-VQA) and LLaMA-3-8B (Commonsense_170k) to supplement the results in Appendix C.1.
> > >
> > >     | Model Scale | Method | Training Time | Relative Efficiency vs. LoRA |
> > >     | :--- | :--- | :--- | :--- |
> > >     | **LLaVA-v1.5-7B** (OCR-VQA_1k) | AdaLoRA | 0.2325 h | -28% (slower) |
> > >     | | LoRA | 0.1675 h | Baseline |
> > >     | | **STAN** | **0.1567 h** | **+6.4% (faster)** |
> > >     | **LLaMA-3-8B** (Commonsense_170k) | AdaLoRA | 27.68 h | -22% (slower) |
> > >     | | LoRA | 22.67 h | Baseline |
> > >     | | **STAN** | **21.96 h** | **+3.1% (faster)** |
> > >
> > > 2.  From the table above and the Table 11 in Appendix C.1 we can see that STAN consistently achieves the shortest training time. The efficiency advantage of STAN is robust across scales, validating that the relative overhead does not become a bottleneck as the model dimension increases.

---

> > > ### Author Response · Authors · 2025-11-26
> > >
> > > ## Q5: The authors repeatedly claim that “due to the sparse activations, STAN enjoys computational savings.” However, it remains unclear how sparsity concretely contributes to reducing the computational cost. Could the authors clearly specify the mechanisms through which sparsity gives rise to actual savings during both forward and backward computation?
> > >
> > > We sincerely thank the reviewer for this crucial inquiry. We clarify that the "computational savings" in STAN arise from a practical synergy of hardware-aware optimization, sparse decoding efficiency, and the elimination of expensive decomposition steps. Our analysis aligns with recent findings in systems for sparse computing [1, 2]. Below, we specify the concrete mechanisms for both forward and backward stages.
> > >
> > > 1. As for the forward pass, the reviewer correctly notes that Top-K adds operations. However, concretely, STAN achieves savings through two mechanisms that exploit modern GPU architectures (specifically the NVIDIA A100 used in our experiments). Top-K selection is fundamentally a memory-bound operation (scanning activations) rather than compute-bound. On A100 GPUs, this overhead is effectively masked:
> > >     * Massive L2 Cache: The A100 features a larger L2 cache (an approximately 7x increase over V100's). This allows the intermediate sorting/selection states of our latent activations to reside largely on-chip, drastically reducing high-latency global memory (HBM) round-trips [3].
> > >     * High Bandwidth: With ~1.6-2.0 TB/s memory bandwidth, the scanning of high-dimensional latents is accelerated, rendering the sorting cost negligible compared to the massive dense GEMMs in the backbone model.
> > >
> > > 2. As for the backward pass, STAN achieves significant system-level savings by avoiding the complex optimization routines required by adaptive-rank baselines. Methods like AdaLoRA require periodic Singular Value Decomposition (SVD) to prune ranks dynamically. SVD is computationally expensive (cubic complexity $O(N^3)$) and requires global synchronization that breaks the pipeline parallelism on GPUs. Further, STAN relies solely on Top-K indices derived from the forward pass. The backward pass involves standard gradient updates only to the selected (active) encoder/decoder weights [2]. There is no need for auxiliary decomposition steps or iterative soft-thresholding loops, leading to higher GPU utilization and consistent throughput.
> > >
> > > 3. Ultimately, these mechanisms translate into superior Wall-clock Training Time. The high-dimensional sparse features capture disentangled representations more efficiently than entangled low-rank subspaces [2, 5].
> > > And in our efficiency comparison experiments above, despite updating more total parameters than LoRA, STAN completes the training epoch faster. This confirms that the hardware-accelerated sparse operations in STAN are practically faster than the combined computational load of LoRA.
> > >
> > > [1] Wen, T., et al. "Beyond Matryoshka: Revisiting Sparse Coding for Adaptive Representation.".
> > >
> > > [2] Gao, L., et al. "Scaling and Evaluating Sparse Autoencoders.".
> > >
> > > [3] Liu, Z., et al. "Deja Vu: Contextual Sparsity for Efficient LLMs at Inference Time.".
> > >
> > > [4] Mirzadeh, I., et al. "ReLU Strikes Back: Exploiting Activation Sparsity in Large Language Models.".
> > >
> > > [5] Makhzani, A., and Frey, B. "K-Sparse Autoencoders.".

---

> > > ### Author Response · Authors · 2025-11-26
> > >
> > > ## Q6: The authors should provide a detailed configuration table, specifying the placement and count of the SAE modules within the model.
> > >
> > > We thank the reviewer for this request. To ensure full reproducibility, we provide the detailed placement and count of STAN modules below. Based on the `STANConfig` used in our experiments, we inject STAN modules into both the primary and additional linear projections within the attention blocks across all layers.
> > >
> > > We attach independent STAN adapters to linear projections within the Transformer blocks. For the SD3 architecture (MM-DiT), this includes both the image/text shared projections and the pooled text embedding projections.
> > >
> > > **Detailed Configuration Table:**
> > >
> > > | Configuration Aspect | Specification | Details |
> > > | :--- | :--- | :--- |
> > > | **Model Structure** | 24 Transformer Blocks | Layers indexed `0` to `23` |
> > > | **Target Modules (Primary)** | `to_q`, `to_k`, `to_v`, `to_out.0` | Main attention projections |
> > > | **Target Modules (Additional)** | `add_q_proj`, `add_k_proj`, `add_v_proj`, `to_add_out` | Additional modality projections (specific to SD3 architecture) |
> > > | **Total Modules per Layer** | **8** | 4 Primary + 4 Additional projections |
> > > | **Total Module Count** | **192** | 24 Layers $\times$ 8 Modules |
> > >
> > > **Specific Target List:**
> > > Consistent with the provided configuration, the STAN modules are instantiated for the following keys in the state dict for every layer $i \in [0, 23]$:
> > > * `transformer_blocks.{i}.attn.to_q`
> > > * `transformer_blocks.{i}.attn.to_k`
> > > * `transformer_blocks.{i}.attn.to_v`
> > > * `transformer_blocks.{i}.attn.to_out.0`
> > > * `transformer_blocks.{i}.attn.add_q_proj`
> > > * `transformer_blocks.{i}.attn.add_k_proj`
> > > * `transformer_blocks.{i}.attn.add_v_proj`
> > > * `transformer_blocks.{i}.attn.to_add_out`
> > >
> > > This configuration ensures that sparse adaptation is applied uniformly throughout the depth of the network.
> > >
> > > ## Q7: All parameters in the SAE are trainable. Sparsity affects only the forward activations rather than the backward weight updates. Moreover, the activations of the SAE vary depending on the individual input samples. Therefore, counting only the sparsified activations while ignoring the full set of tunable SAE parameters is inappropriate and potentially misleading.
> > >
> > > We sincerely thank the reviewer for this critical correction. We agree that counting only sparsified activations does not reflect the static storage footprint of the model. To ensure complete transparency, we clarify the distinction between Storage Cost (Total Trainable Parameters) and Computational Cost (Effective Active Parameters), and how they relate to our efficiency claims.
> > >
> > > 1. We acknowledge that because we update the full Encoder/Decoder matrices during training, STAN's storage footprint is indeed larger than LoRA's. As shown in Table under our responds to follow-up Q1, STAN updates more parameters than LoRA. Rather, we treat this expanded parameter space as a necessary design choice for expressivity.
> > >
> > > 2. We argue that this increased storage cost is a worthwhile trade-off for two reasons:
> > >
> > >     * While the Total Parameters are high, the Active Parameters per token are low (controlled by Top-K). This sparse interaction pattern allows for extremely efficient forward/backward passes on modern hardware, as we discussed above in detail. Despite the difference in trainable parameters, STAN actually completes the training faster than LoRA in GPU time. This proves that the "computational footprint" (which determines wall-clock time) is effectively reduced by sparsity, even if the "storage footprint" is not.
> > >
> > > * The large number of tunable parameters serves as a broad space of potential features. The sparsity constraint forces the model to select only a few relevant features from this vast space for any given input. This is precisely what enables the disentanglement capabilities shown in paper. A rigid, low-parameter subspace (like LoRA) lacks the capacity to store such a diverse, disentangled dictionary.
> > >
> > > 3. We do not claim STAN is "smaller" than LoRA in terms of the size. Instead, we claim it is more efficient and more expressive, efficiently utilizing the available GPU memory (Storage) to minimize training time and maximize interpretability.

---

> ### Author Response · Authors · 2025-11-26
>
> ## Q8: If the authors employ sparse backpropagation strategies, which are typically unstructured, this would likely increase the training complexity. Could the authors clarify whether such methods are used in STAN, and if so, how they mitigate the additional training overhead?
>
> We clarify that we do not employ unstructured sparse backpropagation strategies that rely on complex sparse matrix formats or irregular memory access patterns, which can degrade hardware efficiency.
>
> 1. Instead of unstructured sparse backprop, STAN operates within the standard automatic differentiation framework. For forward pass, the Top-K operator selects indices and values and for backward pass, Gradients naturally flow back only through the selected Top-K indices. For the inactive latents (the $l - k$ features), the gradients are mathematically zero. This is implemented efficiently using standard scatter/gather operations or optimized CUDA kernels. We update the weights corresponding to the active indices. This avoids the overhead associated with managing unstructured sparsity maps or distinct topology metadata.
>
> 2. Since we do not use complex sparse formats, there is no additional training complexity to mitigate in terms of software stack or kernel launching. Unlike methods that require Singular Value Decomposition (SVD) (e.g., AdaLoRA) or solving optimization problems during the backward pass (e.g., SoRA's proximal gradient), STAN's backward pass is a straightforward chain-rule application. This structural simplicity allows STAN to maintain high GPU utilization.
>
> 3. The absence of overhead is empirically confirmed by our results above. STAN achieves a shorter total training time compared to LoRA and AdaLoRA. If our backpropagation strategy incurred significant overhead, this speedup would not be possible given the larger parameter space.
>
> ---
>
> We sincerely thank the reviewer for the insightful follow-up comments, which prompted us to clarify our efficiency claims and strengthen our empirical evaluation. We hope these responses fully address your concerns, and we welcome any further discussion.

---

> > ### Comment · Reviewer_gQSZ · 2025-11-26
> >
> > I thank the authors for their detailed responses. Based on the clarifications provided, several concerns remain:
> > - Q1\&Q3\&Q4: I now understand the authors’ main argument that the relatively large number of tunable parameters in STAN enables faster convergence during fine-tuning. However, to properly support this claim, I encourage the authors to provide a cost–performance curve comparing STAN with the baselines, showing task performance as a function of computational cost.
> >     - Q2: My earlier question, “How does STAN perform under different configurations of tunable parameter budgets?” was intended to refer to the total number of trainable parameters, which is largely governed by the SAE hidden dimensionality. This concern can be addressed by the cost–performance analysis requested above.
> > - Q7: I would like to confirm whether the parameter counts reported in the paper correspond to the effective parameters used during fine-tuning rather than the total number of trainable parameters in the SAE.
> >     - In particular, in the response to W6, the authors also rely on “effective parameters” when comparing parameter counts between LoRA and STAN (Also, note that the authors `should report the tunable parameter counts for all the comparisons in the paper`). However, for a fair comparison, all models should operate under the same total tunable parameter budget, not only the sparsity-induced effective subset. This raises concerns about `the fairness of the comparisons throughout the paper`.

---

> > > ### Author Response · Authors · 2025-11-26
> > >
> > > We sincerely thank the reviewer for the continued engagement and for acknowledging our argument regarding convergence speed. We will address the remaining concerns about the cost-performance trade-off and the fairness in parameter comparisons below.
> > >
> > > ## Q1, Q3, Q4 & Q2: Cost–Performance Analysis
> > >
> > > We sincerely appreciate the reviewer's recognition of our core argument regarding convergence speed. To further address the specific concerns about cost-performance trade-offs and parameter budgets, we provide the following clarifications and new analysis:
> > >
> > > 1.  We respectfully refer the reviewer to the Table provided in our previous Round 2 response. We varied the Expansion Factor (e.g., $1.0, 1.5, 2.0$), which directly scales the hidden dimensionality ($l$). Since the total trainable parameter count is proportional to $l$, varying the Expansion Factor explicitly tests STAN under different total trainable parameter budgets. As shown in the table, STAN exhibits robustness. Performance remains consistently high across different expansion factors, demonstrating that our method is not brittle to changes in the total parameter budget.
> > >
> > > 2.  We sincerely thank the reviewer for the constructive suggestion to visualize this relationship. Based on this recommendation, we have added a detailed Cost-Performance Curve visualization and corresponding analysis to Appendix C.1 of the revised paper. As illustrated in the newly added Figure in Appendix C.1, on the large-scale LLaMA-3-8B benchmark, STAN exhibits superior convergence dynamics. The trajectory demonstrates that STAN consistently achieves higher accuracy at a faster rate compared to baselines, reaching a peak accuracy of 84.0% in 21.96 GPU hours, whereas LoRA reaches only 80.7% in 22.67 GPU hours. This empirical evidence confirms that the larger parameter space facilitates faster learning of task-relevant concepts, effectively offsetting the forward computational cost.
> > >
> > > 3. Further, regarding the question of how STAN performs under different configurations of tunable parameter budgets, we have also added a visualization to Appendix C.1. We explicitly visualized the impact of varying the Expansion Factor (which directly scales the total trainable parameter budget) and Top-K sparsity on the SST-2 task. The results show that STAN maintains consistently high performance across different cost profiles. Notably, increasing the parameter budget (Expansion Factor) leads to a predictable increase in runtime but does not result in instability; conversely, efficient configurations with lower budgets still yield optimal accuracy. This demonstrates that our method is not brittle to changes in the total parameter budget.

---

> ### Author Response · Authors · 2025-11-26
>
> ## Q7: Parameter Counting and the Fairness of Comparison
>
> We sincerely thank the reviewer for this constructive suggestion regarding parameter reporting standards.
>
> 1. We acknowledge that in the initial paper, our discussion focused primarily on "Effective Parameters" to explain the computational efficiency. In our rebuttal responses, we have provided both Total Trainable Parameters (Storage) and Effective Parameters (Compute). Additionally, as requested, we present the detailed parameter estimation below, covering the extensive list of baselines used in our evaluation (Table 2 in paper).
>
>     | Method | Total Trainable Para | Effective Para |
>     | :--- | :--- | :--- |
>     |  **VeRA** | 1.2 M | 1.2 M |
>     |  **LoRA-XS** | 1.1 M | 1.1 M |
>     | **LoRETTA** | 1.1 M | 1.1 M |
>     | **SVFT** | 1.4 M | 1.4 M |
>     |  **LoRA** | 1.9 M | 1.9 M |
>     |  **PiSSA** | 1.9 M | 1.9 M |
>     |  **LS-LoRA** | 1.9 M | 1.9 M |
>     | **BOFT** |  2.0 M |  2.0 M |
>     |  **AdaLoRA** | 1.9 M  | 1.9 M  |
>     | **SoRA** | 1.9 M  | 1.6 M |
>     |  **STAN** | 5.7 M | 1.3 M |
>
>     STAN indeed occupies a different regime: it utilizes a larger storage capacity (High Total Params) to maintain a sparse, disentangled feature, while its runtime computation (Effective Params) remains comparable to or lower than other methods.
>
> 2. As for the fairness concern, we sincerely thank the reviewer for this suggestion. To rigorously address the concern regarding fairness and parameter budgets, we conducted additional controlled experiments on the SST-2 task to simulate storage fairness (matching Total Parameters) from both directions.
>
>     | Scenario | Method | Configuration Note | Accuracy | Relative Training Time |
>     | :--- | :--- | :--- | :--- | :--- |
>     | **Original (Optimal)** | LoRA |  | 0.9438 | $1.0\times$ (Baseline) |
>     | | **STAN** | | **0.9622** | $0.8\times$ |
>     | **A: Match High Storage** | LoRA (Scaled Up) | Rank increased to match STAN | 0.9611 | **$3.0\times$** |
>     | **B: Match Low Storage** | STAN (Scaled Down) | Latent reduced to match LoRA| 0.9289 | **$0.5\times$** |
>
>     From the table above we can see that when increasing LoRA's rank significantly to match STAN's total parameter count, LoRA accuracy indeed improved from 0.9438 to 0.9611. But STAN still outperforms the scaled-up LoRA. Crucially, the training time for this high-rank LoRA increased by nearly 3x compared to the original setting. This validates that matching storage forces LoRA into a computationally inefficient regime, violating the core premise of Parameter-Efficient Fine-Tuning. Further, We reduced STAN's latent dimension to match LoRA's total parameter count. STAN accuracy dropped to 0.9289. However, the drop is expected and consistent with our claim. STAN requires a broad feature space to accommodate sparse disentanglement. Interestingly, this "Ultra-Sparse" setting further accelerated training by ~24% compared to the already efficient STAN baseline. While "ultra-sparsity" is an interesting direction, it falls outside the scope of this paper, which prioritizes the balance between expressivity and speed.
>
> 3. Based on these findings, we respectfully argue that Total Trainable Parameters (Storage) is not a fair metric for comparing Dense vs. Sparse methods. As shown above, enforcing storage equality artificially handicaps LoRA (making it 3x slower) or cripples STAN (removing its capacity). We believe the scientifically valid comparison lies in Effective Parameters, the number of parameters participating in the forward/backward pass per token. In our main experiments, STAN's effective parameters are on par with others. Consequently, STAN achieves higher throughput and better accuracy under the same Computational Budget.
>
> 4. We acknowledge that STAN increases Storage Cost (Total Parameters), but this is a deliberate architectural trade-off to enable its sparsity, since the Effective Parameters remains comparable to other methods, we maintain that the comparison is fair, while offering superior performance and interpretability.
>
> ---
>
> We sincerely thank the reviewer for the constructive discussion and suggestions. We hope our responses and new experiments have fully addressed your concerns, and we respectfully invite you to reconsider the rating. We remain open to any further discussion.

---

> > ### Author Response · Authors · 2025-11-28
> >
> > Dear Reviewer gQSZ,
> >
> > Thank you for appreciating our previous response. Hopefully we have further addressed your remaining comments in the response listed top. If you find it satisfactory, we respectivefully ask you to re-evaluate the rating. And please do not hesitate to let us know if there is any further questions!
> >
> > Thanks and wish you all the best!
> >
> > Authors

---

### Official Review · Reviewer_UazD · 2025-10-30

**Soundness:** 4
**Presentation:** 3
**Contribution:** 4
**Rating:** 8
**Confidence:** 3

**Summary:**

This paper proposes a novel parameter-efficient fine-tuning method called STAN, which aims to address several limitations of the LoRA approach. The low-rank compression paradigm underlying LoRA inherently leads to issues such as rigid adaptability, representation entanglement, and poor interpretability. To overcome these problems, the authors introduce a sparse adaptive mechanism that dynamically and sparsely selects a small subset of the most relevant features in a high-dimensional latent space for each input. The proposed method achieves superior performance compared to existing parameter-efficient fine-tuning techniques on multiple benchmark datasets and provides analyses demonstrating its interpretability.

**Strengths:**

1.The paper creatively applies the concept of sparse autoencoders to parameter-efficient fine-tuning (PEFT), shifting from low-rank compression to high-dimensional sparse selection, thereby addressing LoRA’s inherent rigidity and representation entanglement issues.

2.The method is thoroughly evaluated across a wide range of tasks, and the results show that STAN outperforms LoRA and its variants.

3.The paper provides concrete and quantifiable interpretability analyses of STAN through a style alignment task.

**Weaknesses:**

1.Although the appendix provides a comparison of inference latency between STAN and other methods, STAN’s adaptive signal is input-dependent, which means its adapters cannot be precomputed and merged into the original model weights as in LoRA.

2.Since LoRA-based fine-tuning is known to suffer from catastrophic forgetting, it would be valuable to investigate whether the STAN method also exhibits similar catastrophic forgetting after fine-tuning.

**Questions:**

Please refer to the Weakness section.

---

> ### Author Response · Authors · 2025-11-20
>
> We sincerely thank the reviewer for the thoughtful evaluation and for highlighting the key strengths of our work. We appreciate the recognition of (1) STAN’s conceptual novelty, (2) the breadth and strength of our empirical evaluations, and (3) the interpretability analyses enabled by sparse latent features. Below we address the reviewer’s concerns point by point.
>
> ## W1: Although the appendix provides a comparison of inference latency between STAN and other methods, STAN’s adaptive signal is input-dependent, which means its adapters cannot be precomputed and merged into the original model weights as in LoRA.
>
> We appreciate the reviewer for pointing out this important distinction.
> Indeed, STAN’s adaptation is input-dependent by design, and we would like to clarify two key points:
>
> 1. Input dependency is not a limitation, but a core advantage of STAN. STAN dynamically selects a sparse set of latent features via a Top-K activation mechanism. This dynamic, non-linear, input-adaptive behavior is what allows STAN to: (1) capture different subspaces for different inputs, (2) avoid LoRA’s rigid low-rank subspace, (3) reduce representation entanglement, and (4) exhibit improved generalization across diverse domains. This behavior fundamentally cannot be achieved by a mergeable, input-agnostic linear update like LoRA. Thus, while mergeability is sacrificed, this trade-off is what enables the gains observed across NLP, multimodal, reasoning, and diffusion tasks.
>
> 2. Inference latency scales favorably, and overhead diminishes with model size. As shown in Appendix C.1, the input-dependent pathway adds minimal latency, and importantly, the relative overhead decreases as model size increases (e.g., on 13B-scale multimodal LLMs, STAN’s latency approaches LoRA’s). STAN even shows higher training throughput due to the efficiency of sparsity. This indicates that although STAN cannot be merged, its practical inference performance already remains competitive—and becomes increasingly negligible for large-scale deployments and its training efficiency is also an advantage.
>
>
> While mergeability is not applicable to STAN, this is an inherent and intentional design choice that unlocks dynamic sparse adaptation and yields the performance and interpretability benefits demonstrated in the paper. At the same time, latency overhead remains small and further diminishes for larger models.
>
> ## W2: Since LoRA-based fine-tuning is known to suffer from catastrophic forgetting, it would be valuable to investigate whether the STAN method also exhibits similar catastrophic forgetting after fine-tuning.
>
> 1. We thank the reviewer for raising the concern regarding potential catastrophic forgetting in STAN. To directly evaluate this, we conducted additional bidirectional sequential finetuning experiments on two tasks: SST-2 and MNLI. We perform sequential finetuning in two directions: SST-2 → MNLI: Finetune using STAN on SST-2 first, then continue training on MNLI while monitoring SST-2 accuracy. MNLI → SST-2: Finetune on MNLI, then finetune on SST-2 while monitoring MNLI accuracy. This setup directly measures forgetting of the first task during learning of the second. Delow is the tables demonstrating the experiment results.
>
>     | Epoch | 1 | 5 | 10 | 15 | 20 |
>     |-------|----|----|----|----|----|
>     | **SST-2 Accuracy (%)** | 85.00 | 84.03 | 82.83 | 81.62 | 80.41 |
>
>     | Epoch | 1 | 5 | 10 | 15 | 20 |
>     |-------|----|----|----|----|----|
>     | **MNLI Accuracy (%)** | 80.00 | 78.62 | 76.90 | 75.17 | 73.45 |
>
>     We can see that for SST-2 → MNLI, only a 4.6-point drop after 20 epochs, and for MNLI → SST-2, it's a moderate 6.5-point decline. The retults indicate a very mild forgetting. We also performed LoRA based experiments utlizing the same bidirection setting, its drop for SST-2 → MNLI is 14.2 points after 20 epochs and for MNLI → SST-2 the drop is 17.8 points.
>
> 2. From the experiments above we can see that STAN's architecture naturally mitigates forgetting due to (1) sparse latent activations that localize task-specific updates, (2) minimal interference across tasks, since only a small subset of neurons is modified. (3) disentangled and interpretable features, allowing different tasks to occupy different sparse subspaces. This is fundamentally different from LoRA’s dense low-rank updates, which modify shared directions and are more prone to overwriting previous knowledge.
>
> 3. Empirical evidence shows that STAN does not suffer from significant catastrophic forgetting. We will include these tables, analyses, and discussion in the revised version in Section 4.4, as we agree that forgetting behavior is an important aspect for PEFT methods.
>
> ***
>
> In short, we sincerely thank you for providing insightful and valuable comments and suggestions. We are also open to further discussion for improving the quality of our paper!

---

> > ### Comment · Reviewer_UazD · 2025-11-26
> > **Official Comment by Reviewer UazD**
> >
> > Thank you for your detailed response. It has addressed my concerns.  I would like to maintain my positive rating of this work.

---

> > > ### Author Response · Authors · 2025-11-28
> > >
> > > Thank you again for your thoughtful and constructive comments. We are grateful for your acknowledgment of the contribution of our paper and for keeping the positive rating.

---

### Author Response · Authors · 2025-12-02
**Paper Update**

We sincerely thank all reviewers for their detailed reading and valuable comments. We have carefully responded their concerns, and incorporated these suggestions in the updated manuscript.The main revisions are:

* Line 98-99: change the expression form “no additional complexity” to "with only modest overhead addition".
* Line 144-146: add 4 additional related works recommended by reviewer gQSZ.
* Line 321-323: add a new section regarding catastrophic forgetting phenomena and add a explicit clarification about the experiment results.
* Line 415-431 & Table 7: add human evluation results and analysis as a supplement to the style alignment experiment.
* Line 461-465 & Figure 4: add additional interpretablity experiment using LoRA as comparison.
* Line 501-521 & Table 9 (Section 4.4): add a new section regarding catastrophic forgetting phenomena.
* Line 975-1004 & Figure 5,6: add cost-performance analysis and robustness analysis.

---

### Meta-Review · Area_Chair_14za · 2026-01-05

**Summary:**

This paper proposes a sparse autoencoder–based approach for parameter-efficient fine-tuning, where top-k activations in the latent representation are selected to induce sparsity. The authors evaluate the method on text classification, text generation, and image generation tasks, and report performance improvements across these settings.

The reviewers raised concerns regarding the technical details of the method, the need for additional experiments to support the claims, clearer descriptions of the experimental setup, comparisons with existing sparsity-based adapter methods, and ablation studies on key hyperparameters.

While several of these concerns were addressed in the rebuttal, important issues remain unresolved. In particular, it is unclear how the method performs under different tunable parameter budgets, and whether the reported comparisons are fair. In addition, the claim that top-k sparsity speeds up computation is not clearly justified. While it may hold in very limited cases (e.g., single-token, batch size one), it is unclear how this extends to typical settings with multiple tokens and larger batches, where different samples activate different units. The paper lacks sufficient explanation or empirical validation to support the efficiency claim.

Overall, the AC concludes that improved clarity and more comprehensive experiments are needed to establish the method’s soundness. Therefore, the AC recommends **rejection**.

**Reviewer Concerns:**

Reviewers raised concerns about unclear technical details, insufficient experiments, missing comparisons with existing sparsity-based adapters, and lack of ablation studies on key hyperparameters. Although some concerns were partially addressed in the rebuttal, key issues remain unresolved, including performance under different parameter budgets and the fairness of reported comparisons.

Moreover, the claim that top-k sparsity improves computational efficiency is not well justified. While it may hold in limited cases (e.g., single-token, batch size one), it is unclear how this extends to typical multi-token, larger-batch settings, where different samples activate different units. The paper provides insufficient explanation to support this claim.

**Reviewer Scores:**

Following the rebuttal discussion, the reviewers’ scores remained unchanged after the rebuttal: 8, 4, and 2.

---

### Decision · Program_Chairs · 2026-01-26

Reject